# Research Progress on Characterization and Regulation of Forming Quality in Laser Joining of Metal and Polymer, and Development Trends of Lightweight Automotive Applications

Zhenhong Zhou, Xiangdong Gao * and Yanxi Zhang

Guangdong Provincial Welding Engineering Technology Research Center, Guangdong University of Technology, Guangzhou 510006, China
* Correspondence: gaoxd@gdut.edu.cn

**Abstract:** Metal–polymer hybrid structures have been widely used in research into their lightweight automotive applications, because of their excellent comprehensive properties. As an efficient technology for automatic connection of dissimilar materials, laser joining has great application potential and development value in the field of lightweight automotive design. However, due to the physical and chemical differences between metals and polymers, the formation quality of the hybrid joint is seriously affected by defects, low bonding strength, and poor morphology. Meanwhile, it is difficult to meet the demands for lightweight automobiles by considering only bonding strength as the target for forming quality. Therefore, the technological characteristics of metal–polymer hybrid structures for use in lightweight automotive applications are analyzed, the advantages and problems of laser-joining technology are discussed, and the characterization indexes and regulation measures of forming quality in laser joining are summarized. This paper which provides reference and guidance for reliable forming, intelligent development, and lightweight application of laser joining for polymer–metal hybrid structures.

**Keywords:** metal–polymer; laser joining; forming quality; characterization indexes; regulation measures



## 1. Introduction

With the rapid development of the automobile industry and the continuous increase in the demand for electric vehicles, economic and environmental factors including reduction in the weight of automobiles have attracted attention. The development of lightweight vehicles has become the mainstream direction of automobile development [1,2]. According to research data, with a 10% reduction in vehicle weight, fuel efficiency increases by 6–8% [3]. In addition, the life of the optimized vehicle is extended and the slimmed-down vehicle's ability to withstand impact and crash energy is enhanced, meaning that the lighter the body, the better the vehicle's fuel economy, control stability, and crash safety [4]. At the same time, the development of lightweight automotive technology has accelerated due to the emergence and application of lightweight materials [2]. Among these, polymer and fiber reinforced materials have increasingly been used in lightweight automotive design because of their excellent properties such as low density, high specific strength, corrosion resistance, and heat resistance [5,6]. However, polymers often fail to completely replace metals in industrial applications, especially where high strength properties are required. The metal–polymer hybrid not only improves the overall performance of the structure, but also effectively integrates the advantages of the two materials. It can achieve the reductions in weight and cost that represent the goals of lightweight materials, has been widely used and explored, and is considered to have enormous potential in the automotive field [7]. In addition, compared with thermosetting polymers, thermoplastic polymers and fiber composite materials have the advantages of low requirements in the curing stage, excellent forming performance, strong environmental recovery, high weldability, and large-scale

production. These properties give the hybrid structure of metal and thermoplastic polymer great potential in the large-scale production of lightweight automotive components, and have attracted increasing attention.

In order to fully exploit the advantages of the composite metal–polymer structure, it is necessary to effectively connect the metal and the polymer. The joint mode of the hybrid structure includes manufacturing forming technology and joining forming technology [7]. Manufacturing forming technology can directly realize the integration of metal and polymer hybrid structure formation with high formability, but associated problems include long process cycles, high equipment costs, and poor interface quality, which limit the large-scale application and development of the technology. Traditional connection technologies include mechanical connection and adhesive bonding, which are simple low-cost processes widely used in automotive components. However, the mechanical connection process causes damage to the components and increases structural weight, while adhesive connection requires a long curing time, the adhesive is greatly affected by its environment, and the binding quality is unstable. Both have difficulty meeting the lightweight and high strength requirements of automotive components. At present, induction welding, ultrasonic welding, friction welding, and laser-joining techniques for dissimilar materials have attracted the attention of many scholars [8]. Among these, laser joining has attracted wide attention in the context of metal and polymer connection, due to advantages such as high welding efficiency, high operability, flexibility of process, easily realizable automation, production line flexibility, suitability for large-scale manufacturing, etc. [9,10]. Laser joining has great potential for bonding thermoplastic composites to metal, a technique which is also expected to have direct applications in automotive research involving lightweight materials.

In view of the great application potential of laser-joining technology in the field of industrial lightweight manufacturing, laser joining between metals and thermoplastic polymers has been systematically studied by a number of authors, mainly focusing on the formation process, the bonding mechanism, the strengthening of the bonding, and numerical analysis of the bonding interface, based on experiments [11–13]. However, in laser joining of metals and polymers, the formation of hybrid structure is closely related to the process parameters, and it is easy to produce pores, cracks, discoloration, and other defects during the bonding process, which affect the mechanical properties and forming quality of the joint. In addition, most of the existing studies of metal–polymer laser joining have been limited to the assessment of bonding processes and mechanisms, with little reinvestigation of formation objectives beyond the bonding strength of the junctions. Furthermore, in order to meet industrial requirements, the mechanical properties and forming quality of joints should be regulated and optimized by taking into account their bonding strength, initial morphology, bonding morphology, defects, and forming accuracy. At the same time, the process of lightweight automotive design is a multi-objective optimization process involving structure, materials, connection processes, performance, weight reduction, cost, environmental protection, appearance, etc. In order to promote the application of laser joining in lightweight automotive manufacturing, it is necessary to conduct more comprehensive and in-depth research on the forming quality of hybrid structures.

Therefore, in order to better solve the above problems, improve the forming quality of metal–polymer laser joining, and promote the application of this technology in the lightweight automotive industry, laser-joining technology for use with polymer–metal hybrid structures is discussed in this paper. In this review, the development status and technical characteristics of metal and polymer structures with lightweight automotive uses are summarized, as shown in Figure 1. The advantages, application feasibility, and problems of laser joining technology are analyzed and discussed. The characterization indexes and quantification methods of laser-joining forming quality are summarized, and the optimization and control measures of laser-joining forming quality are introduced, to provide a good foundation for the reliable bonding of polymer–metal hybrid structures. This paper provides references and guidance for research and development in the application of automotive lightweight technology.

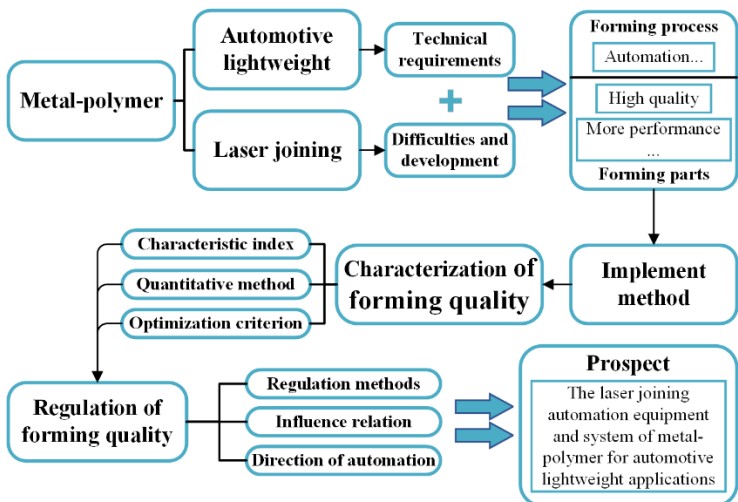

**Figure 1.** An overview of the basic structure of the article.

## 2. Metal and Polymer Hybrid Structure for Automotive Lightweight Materials and Laser Joining

In the field of automotive lightweight materials, it is effective to use the hybrid structure of effectively connected metal and polymer to reduce the weight of components. Among the available methods, laser-joining technology has great development potential as an efficient and automatic process for joining dissimilar materials for use in lightweight automotive applications. However, in order to realize its application and development in lightweight automotive design, it is necessary to consider comprehensively the requirements of the lightweight automotive industry, the basic characteristics of materials, the advantages of laser-joining technology, and technical characteristics, to indicate the direction of development of metal–polymer laser-joining technology.

### 2.1. Research Status of Metal-Polymer Hybrid Structures in Automotive Lightweight

To reduce the weights of automobiles, the introduction of metal–polymer hybrid structures is not a simple component replacement process, but a comprehensive optimization process involving spatial topology, lightweight materials, and connecting technology, based on the different needs of various parts of the automobile.

#### 2.1.1. Automotive Body and Its Structural Parts

The automobile's body and its structural parts, including body frame, body parts, thin-walled tubes, and battery box, comprise its basic spatial structure and main body weight, and are the main objects of lightweight automotive research.

(1) The Automotive Body Frame

The body frame represents the lightweight strategy in macro, requiring construction of a lightweight automobile body through the combination of lightweight material selection and appropriate structure. As shown in Figure 2a, BMW developed the concept of "carbon core", and the steel–aluminum–carbon fiber composite materials were applied to various important structures within the BMW 7 series, such as roof beam, B pillar, C pillar, threshold beam, central channel, and others, reducing the weight by 130 kg compared with the steel component body, with an intensity of 7~9 times that of the traditional steel structure [8,14]. The BMW I3 electric car used a carbon fiber body with an aluminum alloy chassis and body structure bonded by bolts and adhesive bonding, to achieve a lightweight body [15]. Geely's new energy vehicle also used a "steel-aluminum-plastic hybrid body" design [16]. A particular Qiantu Automobile model adopted an aluminum alloy skeleton and carbon fiber covering structure, which were connected by bolts and adhesive bonding to achieve a reduced weight [17]. Considering the complexity of the body frame, some scholars have explored the feasibility of applying polymer and metal multi-combination forms within the body

frame. As shown in Figure 2b, Kopp et al. [18] proposed a vehicle frame structure composed of fiber-reinforced plastic, steel, magnesium, and aluminum materials, with carbon fiber materials applied to the A, B, and C pillars, roof beams, front end structure, and other necessary parts. This was a CFRP (Carbon Fiber Reinforced Composites) intensive multimaterial design, effectively able to reduce weight and cost while ensuring performance. Li et al. [19] designed a sandwich-structure "carbon fiber + aluminum honeycomb + carbon fiber" composite car body bonded by epoxy resin, which not only achieved a light weight, but also greatly improved torsional stiffness of the body.

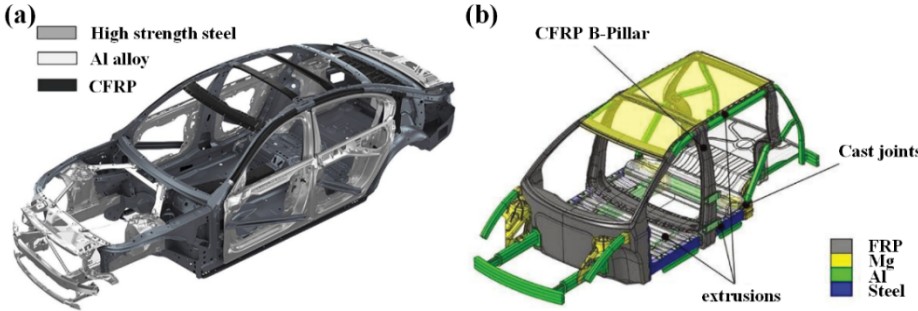

**Figure 2.** (**a**) Metal–CFRP hybrid automobile body of BWM 7-series. Reprinted with permission from [8]. Copyright 2022, Springer. (**b**) Multi-material composite automobile body frame. Reprinted with permission from [18]. Copyright 2012, Elsevier.

(2) The automotive body parts

The automobile body, as the basic frame, is the main research subject for reducing the vehicle's weight. However, the weight-reducing process involving the polymer–metal hybrid structure not only requires combining materials, but also includes comprehensive optimization of materials, structures, and connection technologies based on the performance requirements of components and overall.

Zhu et al. [20] designed a variable section CFRP (epoxy) beam and steel crash box bumper, addressing the aspects of structure optimization, material optimization, and collision safety; the maximum weight reductions were 51.7% and 7.5%, respectively, compared with high-strength steel and uniform CFRP bumper beams. As shown in Figure 3, Chu et al. [21] proposed a multi-objective lightweight evaluation method including cost, material, mechanics, quality, manufacturing, and recyclability, and evaluated the high strength steel (HSS), polymer composite–metal hybrid structure (PMH), and glass fiber reinforced aluminum (GLARE). Yang et al. [22] carried out performance (bending stiffness, torsional stiffness, and modal) analysis of the composite structure hood, and configured a composite structure with PP+EPDM material for the outer plate, LFT (PP) material for the inner plate, and steel internal small parts attached with adhesive bonding. The bending stiffness and torsional stiffness of the composite hood reached 80% of those of the original steel front cover, and the mass weight was reduced by 30%. Deepak et al. [23] proposed a kind of "metal-plastic-metal" sandwich structure automobile hood, based on a material combination of steel, aluminum, and polyurethane foam (epoxy adhesive), and simulated and compared during the driving process the vibration characteristics (aerodynamics, mechanical motion, etc.) of an automobile engine hood with five different material configurations. Park et al. [24], with focus on the stiffness requirements, quality requirements, and production characteristics, combined GFRP (PA6-GF30) with a high-strength steel side plate, constructed the automobile front-end frame using fastening connection, and combined their results with topology optimization to achieve light weight. It can be seen that the design and application of the composite parts involved not simply reverse design or material replacement, but required analysis and design according to the performance of the composite materials, the relevant component characteristics, and the needs of the vehicle body, to adjust and optimize the composite parts and meet the lightweight design needs.

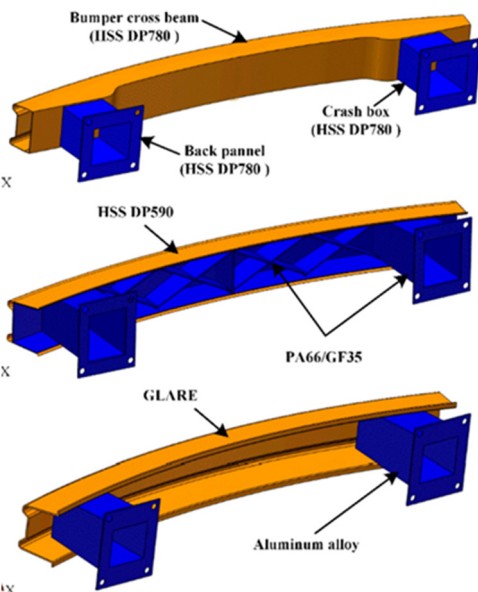

**Figure 3.** Profiles of three bumper systems: HSS system, PMH system, GLARE system. Reprinted with permission from [21]. Copyright 2019, Elsevier.

With the aim of ensuring performance requirements, a variety of lightweight designs can be achieved for similar components by reasonable combination of connection processes, materials, and structures. Porsche has developed a 3D A pillar for its 911 Cabriolet, which combines through adhesive bonding fiberglass mesh panels, honeycomb short-fiber reinforced plastic, and high-strength steel, providing a 5 kg weight reduction over a steel A pillar while maintaining performance [8]. Injection molding technology was applied to bond the steel-fiber composite A column used in high stress lightweight automobile body structure design. The B pillar made using this structure and connection technology reduced the components' weight by 14%, and provided a 25% increase in energy absorption, compared with the traditional B pillar [25]. As shown in Figure 4, Lee et al. [26] conducted a comparative collision test on B-pillar reinforcements made with TWB and CR420/CFRP hybrid composite materials. The collision results showed that the steel–CFRP B pillar fabricated by hot-curing was 10% higher and 44% lighter than the B pillar with TWB. Yang et al. [27] designed a CFRP–TRB hybrid lightweight B pillar by comprehensively considering the side impact performance, structural characteristics, and material characteristics, and optimized the material thickness of each component. The weight of the composite B pillar was reduced by 27.7%, and the side crash resistance was improved. Hoffmann et al. [28] explored the integrated molding of PA6-GF60 and aluminum composite material by examining its application in the cockpit beam.

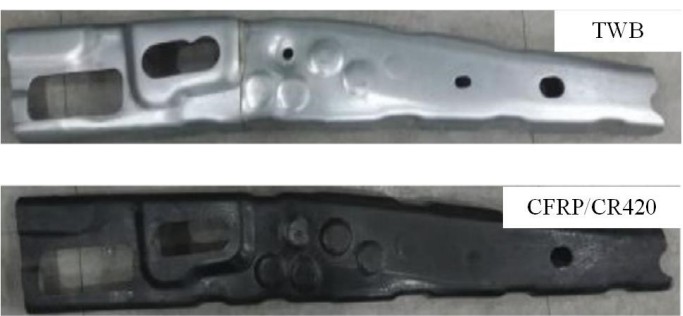

**Figure 4.** B pillars made of CFRP/CR420 composite structure and TWB. Reprinted with permission from [26]. Copyright 2017, Elsevier.

The application of metal and polymer hybrid structures in vehicle bodies is not only limited to the requirements of component performance and weight, but also involves other functional properties of the components, such as assistive enhancement, passive safety, aesthetic comfort, etc. Park and Dang [29], using an injection molding process and considering structural optimization, produced an FRP(PP)–metal composite armrest frame. Compared with the original steel frame, the weight of the optimized composite rear seat armrest was reduced by 50%, and the safety and comfort were improved. The Audi A8 included a carbon fiber rear coiling plate installed with bolts on an aluminum alloy body. The whole panel has uniform force and is easy to disassemble, with a reduction in weight by 50% compared with the raw materials and a 24% increase in torsional stiffness [30]. The Nio ES6 used a light, high-strength carbon fiber plate above the rear axle, which effectively improved the torsional stiffness, passive safety, and durability of the vehicle while reducing weight [31]. As an important part of the car's body, rear doors have been designed with not only the aesthetics, usability, and safety of the car in mind, but also with attention to growing environmental and energy-saving concerns. Ma et al. [32] aiming to meet stiffness, modal, and manufacturing requirements through a multi-step optimization method, changed the materials of the inner and outer panels of the original steel SUV, replacing these with CFRP, and connected the CFRP with steel small parts through adhesive bonding and bolts, to realize the lightweight design of the rear plate. Compared with the metal rear plate, the weight of the optimized composite rear plate was reduced by 37.44%, and the stiffness and first-order modal frequency were improved.

(3) Thin-walled tubes

Thin-walled tubular structures are widely used as energy-absorbing devices to protect against collisions. Thin-walled tubes with metal–polymer structures that meet the requirements of crashworthiness, energy absorption, and light weight are an important research field for the application of energy-absorbing devices in vehicles. From the perspective of cost and performance, Zhu et al. [33] evaluated three types of Al–CFRP (matrix: epoxy resin) composite tubes formed by pressurized thermal curing. The results showed that the cost of the Al (outer)–CFRP (inner) mixed tube was 32.1% lower than that of the pure CFRP tube, the mass was 33.6% lower than that of pure aluminum tube, and the energy absorbed in axial crushing was greater compared with the either of the other composite tubes. These findings can be applied to the automobile's energy absorption box to improve the passive safety of vehicle. Bambach et al. [34,35] explored the feasibility and performance of carbon fiber epoxy composite material–metal (steel/carbon fiber, aluminum/carbon fiber) composite tubes. Ma et al. [36] studied the energy absorption characteristics and application of a CFRP (epoxy resin) and steel hybrid structure using adhesive bonding under transverse load, and proved that the hybrid structure tube could be applied to frames of racing cars. The team also took the car door anti-collision beam as a research object, and explored the lateral bending and energy absorption properties of CFRP (epoxy resin)–aluminum hybrid tubes with adhesive bonding. The results shown that compared with the pure metal anti-collision beam, the CFRP–Al hybrid anti-collision beam reduced the weight by 9.3%, while the load bearing and energy absorption capacity increased by about 7%. In addition, the maximum intrusion and the peak intrusion velocity of the inner decreased panel decreased and the crashworthiness of the door increased under vehicle collision conditions [37,38]. Kim et al. [39] studied the energy absorption performance and bending collapse behavior of Al–CFRP hybrid tubes bonded by co-curing under quasi-static transverse load, and suggested that the hybrid beam could be applied to the lightweight frame of electric vehicles.

(4) Battery box

The battery box consists of an upper cover and a bottom plate. Of these, the bottom plate is the main structural component, bearing the weight of the entire battery pack. The upper cover is merely a cover piece used to protect and seal the entire battery [16]. Therefore, the weight requirements of the battery box need to be approached differently and analyzed as a whole. Schmerler et al. [40] adopted a multi-material hybrid structure consisting of

glass-fiber-reinforced polyamide 6 (GF-PA6), aluminum foam, and solid aluminum, to replace the steel upper cover and achieve an effective combination of strength, corrosion resistance, and a weight reduction of 23%. Based on the idea of "replacing steel with plastic", Liu et al. [41] adopted 3 mm GFRP(PP) to replace 1 mm steel plate for the upper cover of the battery box, and determined the selection of product performance parameters by modal analysis. Liu et al. [42] studied the mechanical properties of a PA66-CF bottom plate, and carried out simulation analysis and performance evaluation of the battery box. The research showed that the weight of the PA66-CF box was reduced by 84% compared to the metal box, and the maximum stress of the box was reduced by 30–50%, under conditions of sudden stops and sharp turns on a bumpy road. Wang et al. [31] tested a lightweight design compared with the original steel battery box, under static and modal conditions. The hybrid structure battery box with SMC upper cover, CFRP (epoxy resin) bottom plate, and aluminum-alloy-reinforced bracket were connected by adhesive bonding and partial riveting. A high weight loss ratio of 46.15% was achieved, and the stiffness, strength, and modal performance of the battery pack were improved.

### 2.1.2. Other Body Components

Metal–polymer hybrid structures have not only been explored for lightweight components in body structures, but also extensively studied for brake, transmission, drive, and suspension components. As shown in Figure 5a, injection molding was used to combine steel and PA6 in the metal–plastic integrated connection brake pedal, which improved the design freedom and also reduced the component's weight while improving its strength [43]. To improve the performance mechanics of a truck's transmission shaft and reduce the weight of the shaft, Yang et al. [44] designed a CFRP drive shaft tube–metal hybrid shaft-head structure bonded with bolts, and combining simulation and experiment validated the application feasibility of the drive shaft. Compared with the metal shaft, the weight of the hybrid structure was reduced by 36%. Lee et al. [45] designed an aluminum–(carbon + glass) fiber epoxy resin hybrid drive shaft, based on the co-curing process. Compared with the traditional steel drive shaft, the mass was reduced by 75% and the torque capacity was increased by 160%. As shown in Figure 5b, Catera et al. [46] employed simulation analysis to examine meshing stiffness and the influence of composite gears bonded by CFRP (epoxy resin) and steel glue. The results showed that the composite gears were expected to improve NVH performance, compared with steel gears of equal quality. Stötzner [47] described the adhesive bonding of high-strength steel plate (DP800) and GFRP (PA6) to construct the transverse control arm of an automobile rear axle. The overall weight was reduced by more than 20%, demonstrating the potential of a lightweight metal transverse control arm with high bearing capacity and reliable failure resistance. Hexcel [48] used its new carbon prepreg technology to enhance the aluminum subframe, improving overall NVH performance using only 500 g of prepreg, and the Al–CFRP hybrid subframe was significantly lighter than the steel version. DYMAG, a British company, demonstrated a carbon fiber–magnesium hybrid wheel consisting of a carbon-fiber wheel mesh and a magnesium brake disc connected by special titanium-plated hardware that reduces the gyrotron effect, makes cars lighter and accelerate faster, and reduces braking distance [49]. Li et al. [50] developed a split structure composite wheel mainly comprising carbon fiber and an aluminum alloy shell core shaft with adhesive bonding, and completed the specific technical parameter validation. Compared with the traditional technology used in automotive wheels, the guaranteed performance of the composite wheel embodied the concept of lightweight efficiency. As shown in Figure 5c, Mitsubishi Rayon and Enkei developed a hybrid wheel that combined forged aluminum with CFRP to reduce the thickness and weight of the aluminum hub, and effectively reduced tire noise and vibration while maintaining strength [51]. Chen [52], using a hot-pressing molding process, used a basalt fiber (BF)-reinforced degradable polylactic acid (PLA) composite material to replace part of the aluminum alloy (5052), designed and manufactured a vehicle control unit shell, which

achieved a weight reduction of 8.9%, high performance, and environmental protection of the interior parts.

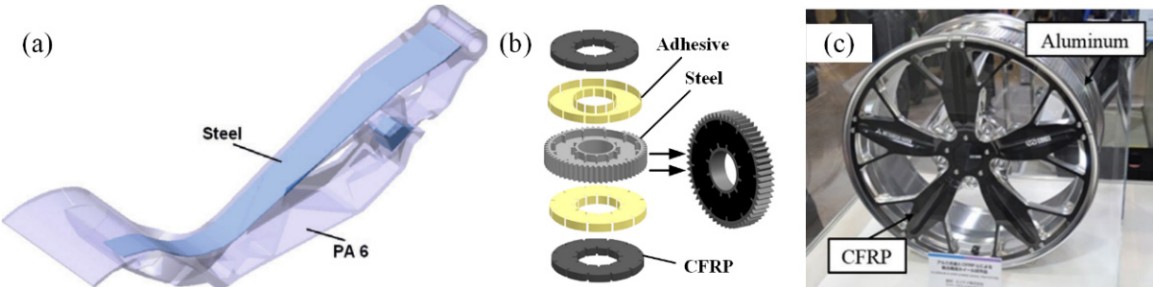

**Figure 5.** (**a**) A hybrid brake pedal. Reprinted with permission from [43]. Copyright 2017, Springer. (**b**) The hybrid gear. Reprinted with permission from [46]. Copyright 2018, Springer. (**c**) The hybrid wheel. Reprinted with permission from [8]. Copyright 2022, Springer.

Table 1 applications of metal–polymer hybrid structures in automobiles, summarizes some existing applications of automotive components. In summary, plastics and their composites with steel, aluminum alloy, magnesium alloy, and other substances have become a popular trend in the development of lightweight automotive materials. The use of metal–polymer hybrid structures for lightweight automotive applications is a multi-objective optimization process, and it involves assessment of the optimal structure, material combination, and advanced connection technologies of hybrid structures, considering the comprehensive requirements of the project.

**Table 1.** Applications of metal–polymer hybrid structures in automobiles.

| Applications | Sources | Materials and Structures | Connection Process |
|---|---|---|---|
| Body frame | BMW 7 series [8,14] | Steel–Aluminum–CFRP | — |
| | BMW-i3 [15] | CFRP cockpit + Aluminum alloy chassis | Adhesive bonding and bolted connection |
| | QIANTU-K50 [17] | Aluminum alloy body frame + CFRP outer covering | Adhesive bonding and bolted connection |
| | Li et al. [19] | Sandwich: Carbon fiber + Al honeycomb + Carbon fiber | Adhesive bonding |
| Bumper | Chu et al. [21] | HSS-DP590+PA66-GF35 hybrid structure | Injection molding |
| | | GLARE (die casting) beam + Aluminum alloy collision box | Adhesive bonding |
| Hood | Ishak [53] | NFRP (PP)+5052 Aluminum alloy laminate | Adhesive bonding |
| | Yang [22] | Outer panel PP+EPDM, Inner panel LFT (PP), Steel widgets | Adhesive bonding |
| | Deepak et al. [23] | Three kinds of sandwich structure: Steel–polyurethane foam–steel, aluminum–polyurethane foam–aluminum, aluminum–polyurethane foam–steel | Adhesive bonding |
| Front-end module | Park et al. [24] | PA6-GF30+ High strength steel side panel | Mechanical fastening |
| | Audi A3 [54] | Steel plate +GFRP (PA6-GF30) | Injection molding |
| | Town&Country [54] | GFRP + Steel plate | Riveting and adhesive bonding |

Table 1. *Cont.*

| Applications | Sources | Materials and Structures | Connection Process |
|---|---|---|---|
| A pillar | Porsche 911 [8] | Fiberglass mesh, honeycomb structure short FRP and high strength steel composite | Adhesive bonding |
| | Drössler et al. [25] | Steel–FRP | Injection molding |
| B pillar | Drössler et al. [25] | Steel–FRP | Injection molding |
| | BMW-7 series [27] | Steel–CFRP | Molded connection |
| | Lee et al. [26] | CR420 steel + CFRP (thermosetting) composite | Thermal curing |
| | Yan et al. [55] | Sheet metal parts + continuous CFRP (epoxy resin) | Adhesive bonding |
| Cockpit beam | Hoffmann et al. [28] | PA6-GF60+Al | Injection molding |
| Door hinge | Yu and Kim [56] | CFRP and 6061 aluminum alloy combined with steel | — |
| Armrest | Park and Dang [29] | Short GFRP(PP) handrail frame + Metal pin | Injection molding |
| Coaming | Audi A8 [30] | CFRP rear coaming + Aluminum body | Bolted connection |
| Car floor | Nio ES6 [31] | CFRP rear floor (embedded in all-aluminum body) | — |
| Rear door | Ma et al. [32] | SUV CFRP rear door + Steel widgets | Adhesive bonding and bolted connection |
| Thin-walled tubes | Zhu et al. [33] | Al (outer)–CFRP (inner, epoxy resin as matrix) | Pressure heat curing |
| | Bambach et al. [34,35] | Carbon fiber wound metal (steel/carbon fiber, aluminum/carbon fiber) epoxy resin composite | Fiber winding and adhesive bonding |
| | Ma et al. [36] | CFRP (epoxy resin) + Steel | Adhesive bonding |
| | Sun et al. [37,38] | CFRP (epoxy resin) + Aluminum | Adhesive bonding |
| | Kim et al. [39] | CFRP (epoxy resin) + Aluminum | Co-curing |
| Battery box | Schmerler et al. [40] | Upper cover: GFRP (PA6) + Al foam + Solid Al | Heat curing |
| | Wang et al. [31] | Upper SMC+ Lower CFRP + Aluminum alloy reinforced stent | Adhesive bonding and riveting |
| Brake pedal | Miklavec et al. [43] | Steel (main frame) + PA6 | Injection molding |
| Drive shaft | Yang et al. [44] | CFRP drive shaft tube + Metal shaft head | Bolted connection |
| | Lee et al. [45] | Aluminum + (carbon + glass) FRP (epoxy) | Co-curing |
| Car gear | Catera et al. [46] | CFRP (epoxy resin) + Steel | Adhesive bonding |
| Control arm | Stötzner [47] | High strength steel plate (DP800) + GFRP (PA6) | Adhesive bonding |
| Subframe | Hexcel [48] | Aluminum alloy +FRP | Curing |
| Wheel | DYMAG [49] | Carbon fiber wheel net + Magnesium brake disc | Mechanical joining |
| | Li et al. [50] | Split structure: CFRP shell + Cast aluminum alloy mandrel | Adhesive bonding |
| Interior part | Chen [52] | Control unit housing: BFRP (PLA) + Aluminum alloy | Heat pressing molding |

### 2.2. Basic Characteristics of Metal-Polymer Hybrid Structure

In the use of lightweight metal–polymer hybrid structures, the metal materials include mainly steel, aluminum, magnesium, titanium, and other alloys, while the polymers are generally plastic and composite materials. Among these, plastic (also known as polymer) is a polymer material based on synthetic resin, with different additives and produced under the action of particular temperatures and pressures, molded into a variety of shapes and products. However, plastic has limited thermal resistance and mechanical properties. Therefore, it can only be used after it has been modified into high performance polymer with reinforcing material. Fiber-reinforced thermoplastics have received much attention due to their excellent mechanical properties [57].

In FRP, the fiber functions as the reinforcing element and the polymer as the matrix. The mechanical properties of FRP mainly depend on the content, type, and shape of the reinforcing material [58]. Common types of reinforcement materials include carbon fiber, glass fiber, aramid fiber, basalt fiber, PE fiber, etc. [59–62]. The fiber morphology is divided into discontinuous and continuous fibers. Discontinuous fibers are divided into short fibers, long fibers, and short cut strand felt. Continuous fibers include continuous felt, unidirectional or multiaxial laminates, knitted fabrics, and woven fabrics. In terms of fiber length, there are continuous fibers, long fibers, and short fibers. Polymer matrices can be divided into thermoplastic and thermosetting polymers. Thermoplastic polymers are plastics that are insoluble after curing by heat and cannot be repeatedly softened and molded upon reheating. The most common thermoset resin systems are polyester, ethylene ester, epoxy, phenolic, etc. [63–65]. Thermoplastic polymers, such as polypropylene (PP), polyethylene (PE), polyamide (PA), polyether ether ketone (PEEK), polycarbonate (PC), etc., are plastics that can be heated to a soft flow, then cooled to solidification and physical hardening, while the process can be repeated [58]. Compared with thermoset plastics, thermoplastic polymers have received more attention due to their low curing stage requirements, excellent forming properties, recyclability, etc. [66–71]. However, thermosetting polymers continue to be widely used because of their low cost and good thermal stability [8]. With the development of advanced connection technologies and production technologies for thermoplastic polymer materials, in view of their characteristics, metal–thermoplastic polymer composite structures will continue to have development potential in industrial production, and comprehensive application in lightweight materials for the automotive industry.

### 2.3. The Fabrication Process of Metal-Polymer Components and the Advantages of Laser Joining

Polymer–metal connection processes can be divided into manufacturing forming technology and joining technology. In manufacturing molding, metal and polymer connection technology includes in situ molding (IM: injection molding, 3D printing, etc.) [9] and co-curing forming processes [72,73]. IM, as one of the technologies that can realize connection in a short period of time, has the advantages of high automation, low waste, mass production, convenient recycling, etc. However, it is a discontinuous process that requires significant investment, and the necessary tools and IM machines are expensive, making it difficult to use widely [9,74,75]. The 3D printing method (FDM, fused deposition modeling) is based on heating and melting, depositing layers of polymer filaments onto a metal heating bed to achieve hybridization [9]. Advantages of FDM technology include good cost performance, high operability, and the ability to manufacture complex 3D structures directly, but the forming accuracy is poor, the forming process is slow, and the time consumed is proportional to the model complexity and layer height [76]. Co-curing forming has the advantages of high formability, high form efficiency, and high flexibility. It can fabricate metal–polymer composite components with complex geometries and is widely used in the automotive industry for hybrid structure formation processes [8]. However, with the narrow window of forming process, it is easily affected by the process parameters of fiber and resin base forming, as well as wrinkling, delamination, fiber fracture, and resin extrusion at the joint after curing. Thus, the large-scale application of this process in the forming of automotive metal and polymer hybrid structures is restricted [77,78].



The traditional bonding technologies for connecting polymers and metals have mainly included adhesive bonding and mechanical fastening [79–82]. The adhesive joining process realizes continuous and large-area fastening through the adsorption force of "adhesives" between substrates, with the advantages of insignificant weight increase, uniform stress distribution, and high strength-to-weight ratio, which make adhesive joining better than mechanical fastening in many cases [83–85]. However, this process is inefficient and requires professional workers, a long curing time, and surface treatment, increasing process costs, production time, and environmental impact [83,86]. In addition, adhesives are subject to environmental influences such as humidity and temperature, and subsequently age and weaken joints, resulting in greater long-term uncertainty of structural integrity [87,88]. Mechanical fastening methods include bolting and riveting, mainly through the use of additional clamps (screws, rivets), to form an effective joint between heterogeneous materials, so that metal and polymer materials which have large differences in physical properties can achieve tight connection. Because it involves no additional heat source, the mechanical fastening process can effectively avoid the problems caused by heat input which include joint softening and hard and brittle phase formation at the interface. Furthermore, the mechanical fastening method does not require surface pre-processing and the process is simple and easy to automate, thus it is the most commonly used method for fastening components. However, no matter which riveting method is used to join the polymer to the metal, it inevitably causes damage to the low ductility parent material (especially FRP), which affects the connection strength [8]. Mechanical fastening usually has some drawbacks, such as stress concentration, the need to drill holes, and a long connection time using external fasteners. It is difficult to make holes in composite materials for mechanical connections, and it is easy to produce defects such as delamination and fiber pull-out, which affect the mechanical properties of the structure [89–91]. Moreover, severe tool wear also leads to high machining costs and low efficiency.

The above problems indicate that traditional bonding processes have difficulty meeting the increasing engineering application requirements of hybrid structures. Welding is an important fundamental processing method for the connection of structural parts. The method can include ultrasonic welding, friction stir welding, induction welding, laser welding, etc. It has advantages for joining thermoplastic composite materials and metal heterostructures. Among the various types of welding, ultrasonic welding has the advantages of high joint strength, a short welding cycle, low cost, and ease of controlling the bonding area. However, the welding process is accompanied by melting polymer extrusion, which leads to material deformation and joint strength reduction, and the shape and connection size of heterogeneous components are limited by the size of the ultrasonic indenter [92,93]. Friction stir welding has the characteristics of a short process cycle, no additional material requirements, and high efficiency, but is limited by the thickness of the metal-based material, high energy dissociation, high equipment requirements, and difficulty controlling the melting area, while the joint is prone to pores and other defects [94,95]. Induction welding enables contactless connection of heterostructures, and is cheap and fast. Its main drawback is that there must be sufficient space around the sample for the induction coil, and the size of the induction coil limits the interface connection area [96,97]. With controllable energy, maneuverability, ease of shaping, lack of contact, and ease of automation, laser welding can not only meet the requirements of various technologies and occasions, but also enables large-scale automated fabrication processes. Laser connection technology can weld micro, refractory, non-conductive, or non-magnetic materials, making it suitable for the connection of various joint forms; furthermore, the joint produced is firm, and the size of the joint can change freely [98–101]. In particular, its great potential for use with thermoplastic composite materials and metal heterostructures has made laser welding a research hot-spot in recent years [9,12]. Automotive structures are characterized by a wide variety of components, complex local geometric characteristics, small radius of curvature, and other factors. The laser-joining process is more flexible and more easily achieves local

reinforcement, while its geometric accuracy and forming consistency are high, providing broad prospects for the development and application of automotive hybrid structures.

*2.4. Basic Characteristics of Laser Joining Metal and Polymer*

In the laser-joining process of metal and polymer, the laser energy is absorbed by the metal, then is conducted to the polymer and melts it at the interface [11,102,103]. As the temperature rises further, parts of the polymer decompose and form bubbles in the molten plastic. This activated high temperature polymer melts in full contact with the metal surface, under the combined action of high pressure caused by the generation and expansion of bubbles and the external clamping pressure, to form a strong binding joint after cooling. These bubbles remain in the solidified plastic area. The whole connection process is highly dependent on the fastening device or clip, and it is necessary to ensure that the interface has no air gaps and is in fully contact [9]. Because of the polymer melting process, thermoplastic polymer is the best choice. The process can be classified as laser transmission joining (LTJ) or heat conduction joining (CJ), according to the mode of laser irradiation. The main difference between the two is that when LTJ is used, the laser irradiation directly penetrates the polymer (when the transparency exceeds 60%) to heat the metal surface at the bonding interface; meanwhile, the CJ metal polymer, also known as laser assisted metal and plastic (LAMP), involves a laser directly illuminated on the metal surface of the metal–polymer hybrid structure. For LTJ, the effect is related not only to the transmittance of the polymer substrate, but also to the laser wavelength (laser type) [104,105]. In the whole laser-joining process, the bonding temperature at the interface affects the forming process, and the key parameters of laser joining, including laser power, welding speed, spot area (defocus), and clamping pressure directly control the temperature field of the bonding interface [11]. At certain temperatures, the bonding of metals and polymers depends on the bonding mechanism at the interface. Interfacial bonding mechanisms include mechanical, chemical, and physical bonding. Mechanical bonding refers to the embedding of composite materials onto the concave and convex surfaces of metals, to form mechanical anchorage at the micro level, which is essentially a friction force whose strength depends on the size and density of the microstructure and the surface treatment technique [106–109]. Chemical bonding refers to the formation of new chemical bonds between metal surface atoms and thermoplastic resin surface-active functional groups, due to the mutual migration of charges, thus forming a connection [110,111]. The existence of this bonding mechanism depends on the chemical reaction between the metal and the polymer and the formation of chemical bonds. Physical bonding refers to physical adsorption between the two materials, mainly including van der Waals force and hydrogen bonding. The strength of the bonding is related to the proximity of the material surface and whether or not the electronegative atoms on the material surface share protons [12,112]. With the progress of laser joining of hybrid joints, due to large difference of physical and chemical properties between base materials and the faster melting process of laser processing, the joint is prone to lack of fusion, bubbles (pores), polymer thermal decomposition defects, tunnelling, discoloration, cracks, and other defects, which affect the joint quality [113–116].

In summary, metal–polymer laser joining is a process closely related to the selected laser welding process, laser performance, laser process parameters, and physical, thermal, optical, and chemical properties of metal and polymer materials. The reasonableness of the selected parameters and process is directly reflected in the forming quality. Therefore, to obtain structural parts with good forming quality, it is necessary to comprehensively consider the forming quality characterization and the forming quality control method.

## 3. Forming Characterization of Metal–Polymer Hybrid Joints

The quality of laser joining between metal and polymer is directly reflected in the macroscopic characteristics of the bonding quality. Through the selection, extraction, and quantification of the macroscopic indexes of laser-bonding quality between metal and polymer, the bonding quality of hybrid components can be evaluated intuitively, efficiently,

and in real time, aiming at the goal of lightweight technology, and providing a reference for the accurate control of forming quality and process optimization. Characterization indexes of metal and polymer laser-joining forming quality mainly include weld characteristics of the metal surface, weld characteristics of the bonding zone, mechanical properties, defect characteristics, and sensing-signal characteristics.

### 3.1. Weld Characteristics of Metal Surface

In heat-conduction joining, due to the direct irradiation of the metal surface by the laser, weld seams often appear on the metal surface and subsequently affect the overall formation quality. Therefore, many scholars have studied weld seams on the metal surface and defined them as thermal defects to characterize their influence on the quality of the form.

As shown in Figure 6, Sheng et al. [117] studied laser joining of CFRP with stainless steel and found that the surface of the stainless steel was affected by laser heat, and the thermal defect area was divided into the melting zone (approximately conical) and the heat-affected zone (approximately semi-ellipsoid), and that such thermal defects would change the microstructure and strength characteristics of the stainless steel.

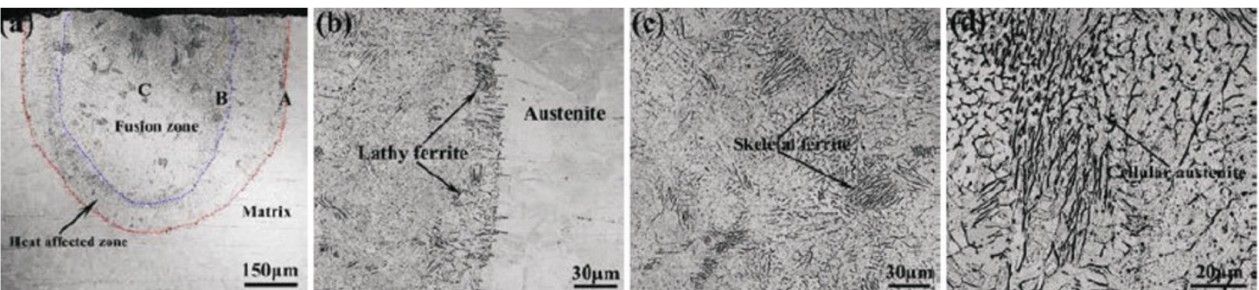

**Figure 6.** (**a**) Macrograph of the laser-scanned stainless steel; (**b**) Microstructure of the stainless steel along the interface of matrix and HAZ (A position); (**c**) Microstructure of the stainless steel adjacent to the fusion zone (B position); (**d**) Microstructure of the stainless steel in the fusion zone (C position). Reprinted with permission from [117]. Copyright 2018, Springer.

Jiao et al. [118] calculated the weld depth on the aluminum alloy surface and the weld depth of CFRTP–Al alloy joints with LAMP, based on numerical simulation. Compared with the experimental results, it was shown that the weld size on the aluminum alloy surface was correlated with the hybrid joint size in the interface. In addition, the authors studied the CJ of CFRTP and stainless steel and found that the CFRTP–stainless steel interface was tightly bonded with no macroscopic cracks or defects of the joint, but that these occurred at the surface where it was affected by the apparent heat of the stainless steel. In this heat-affected region, a large quantity of slag and oxide appeared, the organization and structural characteristics of the stainless steel changed, and the melting zone was lower than the parent metal hardness. It was indicated that the hot defect area should be reduced, on the premise that CFRTP and stainless steel could be combined together in the welding process, the size of the hot defect area could be judged by the width and depth melting, and the forming quality could be characterized, as shown in Figure 7 [119].

It can be seen from the above that the geometric dimension of the weld on the metal surface is related to the joint size and forming quality. However, not all forming qualities can be characterized by metal surface weld seams. As shown in Figure 8, Jung et al. successfully realized the reliable connection of CFRTP–aluminum alloy [106], CFRTP–galvanized steel [112], and CFRTP–stainless steel [120] through high power continuous laser heat-conduction joining. Among their samples, the stainless steel welding process used a disk laser, and the others used a diode linear laser, the stainless steel plate was 3 mm thick, and the rest were 1 mm thick. The results showed that there were narrow welds on the surface of the stainless steel, while there were no welds on the top surface of the aluminum alloy plate, which greatly improved the product quality and reduced

the post-treatment process necessary for the aluminum alloy surface under the LAMP connection. Although there was no weld on the top surface of the galvanized steel plate, the phase transformation rate of the galvanized steel plate coating in the laser-irradiation heating area was increased, which leaded to a slight change in the surface color of the irradiation area compared with the substrate. Overall, the morphology of the surface weld depends on the surface roughness, the thermal physical properties of the materials, and the laser absorption rate.

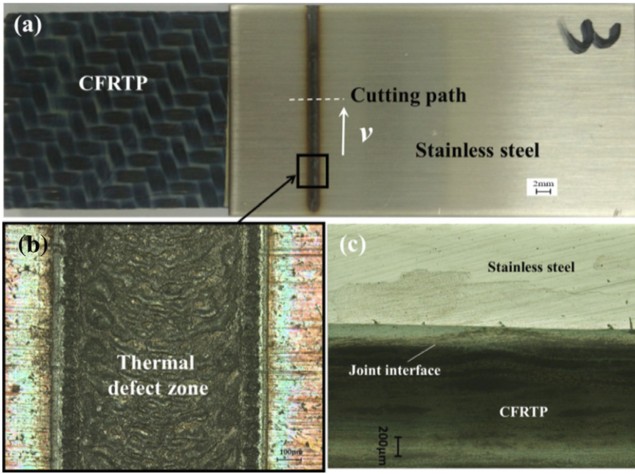

**Figure 7.** (**a**) The front surface of the joint sample; (**b**) The thermal defect zone in the stainless steel surface; (**c**) Cross-sectional photo of the joint. Reprinted with permission from [119]. Copyright 2018, Elsevier.

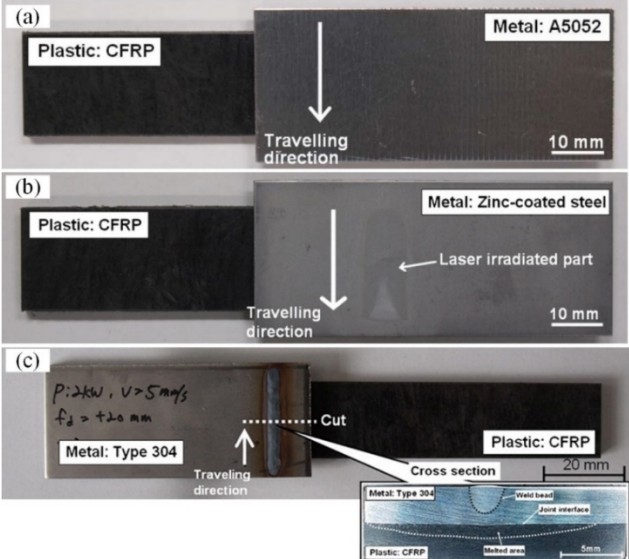

**Figure 8.** Surface morphology characteristics of laser heat-conduction joining of different metal materials: (**a**) CFRTP–aluminum alloy, (**b**) CFRTP–galvanized steel, (**c**) CFRTP–stainless steel. Reprinted with permission from [11]. Copyright 2022, Elsevier.

In summary, during laser heat-conduction joining, thermal defects appear on the upper surface of the metal under laser irradiation, and the degree of thermal influence is related to the technological conditions. The process requires minimizing the size of the heat-affected zone on the premise of ensuring the effective connection of the base metal, and the forming quality is characterized by the geometric dimensions of the weld (such as weld width and depth), or its color, etc.

*3.2. Weld Characteristics in Bonding Zone*

In laser joining, the metal–polymer joint size is closely related to joint strength and forming quality [121,122]. The common weld size parameters of the bonding zone include weld-pool depth (or height), weld-pool width, and bonding area.

3.2.1. Weld-Pool Depth (or Height) and Width

Yusof et al. [123] used Nd:YAG laser spot welding A5052-PET and SUS304-PET to study the influence of weld-pool depth on joint strength, and found that the joint strength was proportional to the depth of the weld pool, and an increase (or decrease) of weld-pool depth led to an increase (or decrease) of joint strength. In addition, they found that the depth of the weld pool was closely related to the welding parameters, and the formation of the weld pool depended on the heat input and distribution. The deeper the weld pool, the stronger the resistance to shear failure, and the stronger the joint [121,124]. Ai et al. [125] carried out experimental research and numerical simulation on LTJ of polyethylene terephthalate (PET) and Ti6Al4V, evaluated the weld geometry and welding quality, and analyzed the weld geometry, weld pool, fluid flow, and pore formation. The results showed that the weld geometry (weld width, weld height) reflected the forming quality of the bonding zone and the influence of process parameters. At the same time, the simulation results showed that the heat absorbed by the laser in the interaction zone was carried by the circulating fluid to the edge of the molten pool, which greatly enlarged the width of the molten pool, and thus played an important role in heat transfer, the geometry of the molten pool, and the formation of the weld. Weld width and weld depth (or height) are key to weld shape size, and can effectively characterize the effects of process parameters on bonding quality. Weld width, which determines processing efficiency and connection stability, is closely related to bond strength, and is the principal indicator of bond quality. In addition, a wider weld facilitates load-bearing applications [126]. Higher weld strength can be obtained by enlarging the weld width, but a weld width that is too large can easily cause high temperatures in the central area, leading to thermal decomposition of the polymer, which then affects the forming quality [127,128]. Therefore, the weld width can be measured to characterize the forming quality.

However, the weld morphology differs in the bonding zones of different laser-joining modes (CJ and LTJ), leading to differences in the calibration of weld width. Meanwhile, the weld width is not equal everywhere, as shown in Figure 9 [129]. Therefore, the average weld width is often taken. In addition, polymers can be transparent or opaque. When a transparent polymer is selected for welding, the weld width of the binding zone can be directly calibrated using the transparent plastic. For these materials, when the width distribution of the joint area is more balanced, the weld width can be observed by taking at least three measurements at different positions along the weld and calculating the mean value [66]. When the width of the binding area is unevenly distributed, the average bond width W (calculated as the ratio of the bonded area to the bond length) can be used [115].

For opaque plastics, the width of the weld section is often extracted by calibration, and the size is easily affected by the excision position. Therefore, to describe reasonably the weld width of opaque plastic in the joint zone, scholars have calculated the joint width from the joint fracture morphology, characterized the forming quality, and systematically analyzed the influence of process parameters on the joint-bonding quality. Jiao et al. [119] carried out experiments using different welding parameters in the fiber laser welding system, and investigated the relationship between welding parameters and the bonding quality of CFRTP–stainless steel, by observing the joint's molten width Wc at the joint fracture after tensile testing. Tan et al. [130,131] studied the laser-joining process of CFRP–titanium alloy at different scanning speeds and defocus distances, and divided the joint fracture morphology into two regions: the bonding region (center) and the resin-bonding region. The width of the resin-bonding region was measured, and the bonding ratio of the carbon fiber and resin mixture in the bonding region was measured by Image J, to study the bonding quality, as shown in Figure 10. The results showed that with the increase of

scanning speed, the bonding width of the resin and the bonding ratio of the carbon fiber to resin mixture first increased and then decreased. With the increase of laser beam diameter, the width of the resin-bonding zone and the bonding ratio of the carbon fiber to resin mixture increased, and the changes of the two corresponded to the fluctuation of fracture load. It was also indicated that the bonding strengths of specimens that had undergone laser bonding at different scanning speeds or defocus values were mainly related to the resin-bonding widths and the bonding ratios between the carbon fiber and the resin mixture in the bonding regions.

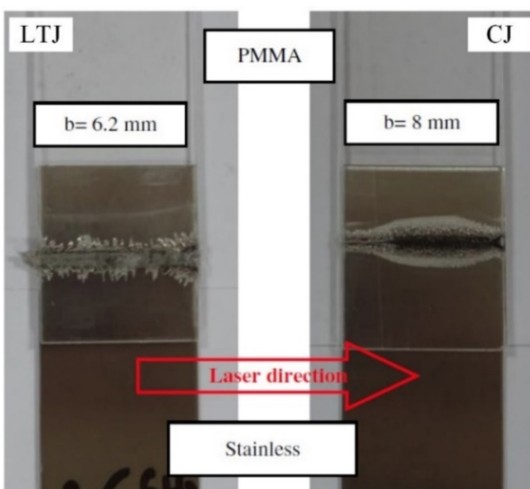

**Figure 9.** Typical joint appearances. Reprinted with permission from [129]. Copyright 2012, Elsevier.

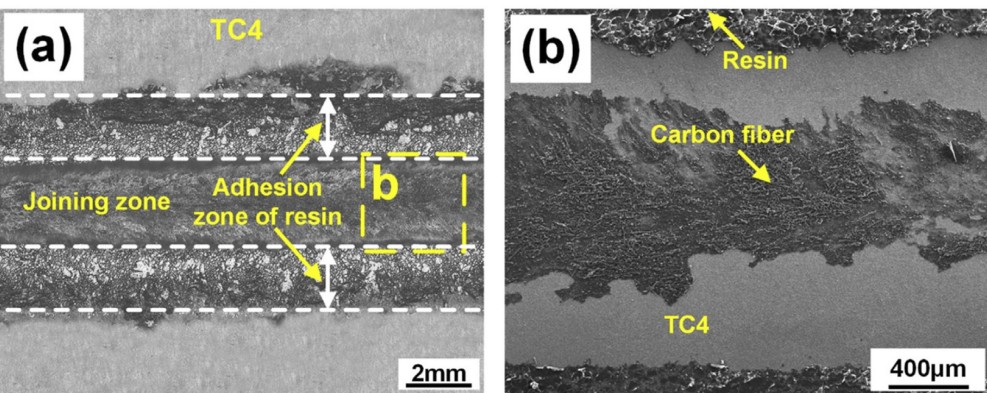

**Figure 10.** Characteristic areas in fracture surface of joint between TC4 and CFRP–PEEK: (**a**) fracture surfaces of the typical joint on the TC4 side, (**b**) detail of joining zone. Reprinted with permission from [131]. Copyright 2019, Elsevier.

### 3.2.2. The Bonding Area

The weld width has been widely used to characterize the forming quality of joints in the bonding region. However, during laser joining of the polymer to the metal, heat accumulation occurs in the bonding region, which causes the welding area to increase with the increase of heat input. The increased heat input and heat accumulation promote the thermal decomposition of the polymers in the bonding zone, aggravate the generation of bubbles, and lead to the appearance of an unconnected area, which distorts the extracted weld-width value, thereby affecting the evaluation of connection strength and quality [124,132]. Therefore, some scholars have used characterization of the effective bonding region, which can better measure the actual bonding status of the bonding area, to evaluate the forming quality.

Pagano et al. [121] explored the feasibility of polylactic acid and aluminum thin films connection by LTJ. By analyzing the morphology of the welding area, the bonded area can

be divided into the molten area and polymer ablative thermal degradation area, represented by molten width ($W_W$) and ablative area width ($W_D$), respectively. The effective bonding area of the joint was given as follows: bonded area = $((W_W - W_D)/2) * I$ (where I is the weld length). The forming quality of the joint was characterized, as shown in Figure 11. The results showed that the process window for reliable welding was very narrow, while the tensile strength of the joint was proportional to the effective bonding area and reached a satisfactory value. It can be seen that the effective bonding area is closely related to joint strength and forming quality.

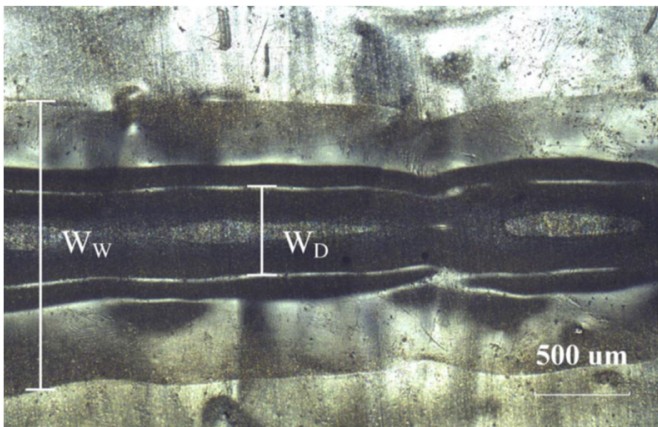

**Figure 11.** Example of an optical micrograph of the welding zone. Reprinted with permission from [121]. Copyright 2016, Elsevier.

However, with respect to the LTJ technique between plastic and metal, the manual calibration of the effective weld joint area is often time-consuming, laborious, and imprecise, resulting in errors in determining its relation to the process parameters. Therefore, the effective bonding size of the bonding area can be directly measured by the vision sensor. Huang et al. [133] studied the LAMP of SUS304-PMMA, measured the weld joint boundary (Sj) by Image J software, and divided the welding region according to color characteristics into effective joint area (Se) and discolored area (Sd), then the effective joint area was calculated to evaluate accurately the shear strength of the joint. Chan and Smith [134] studied LTJ of polyethylene terephthalate (PET) and commercial pure titanium (Ti), and proposed a reliable and quantitative method to calculate the contact area of the bonding region. As shown in Figure 12, the total joint area (TJA) between PET and Ti was divided into the contact area (CA) and non-contact area (NCA). The NCA was the interface area occupied by bubbles, while the CA was subdivided into the clean area (CZ) and the discoloration area (DZ). The surface morphology characteristics of the joint interface were collected by a non-contact laser profilometer, and the contact area (CA) was directly quantified and calculated based on Image J graphic analysis. The joint strength and bonding quality were then characterized, the influence of process parameters on forming quality was analyzed, and the relationship was evaluated between discoloration and joint strength. However, this method has limitations. For shallow bubbles (height less than 60 μm), manual extraction is necessary.

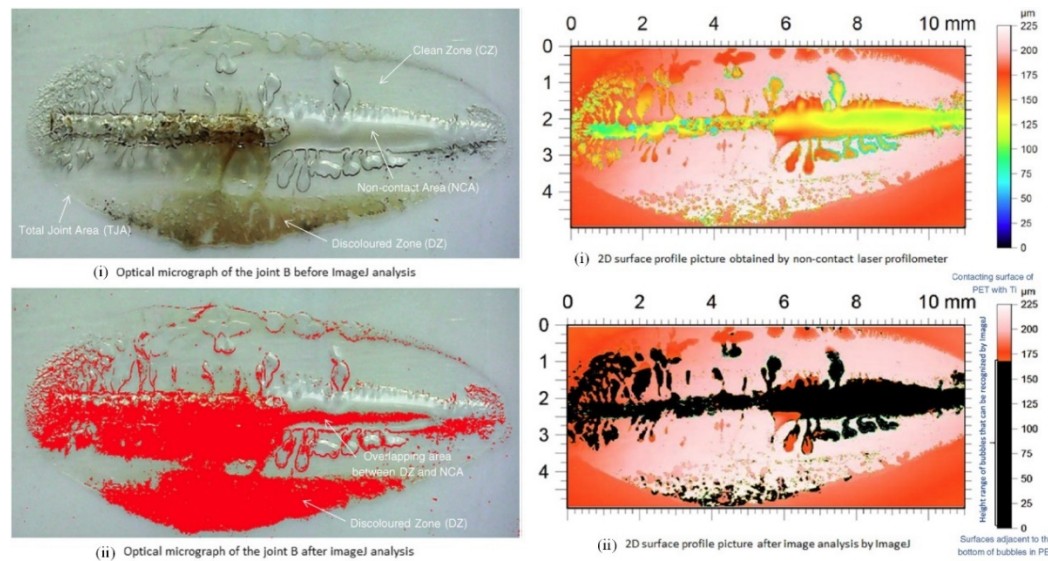

**Figure 12.** The characteristics and quantification of the interface effective contact area. Reprinted with permission from [134]. Copyright 2016, Elsevier.

*3.3. Mechanical Properties*

### 3.3.1. Bonding Strength

Bonding strength is an important standard for evaluating the weld quality of polymer–metal hybrid structures, and it is an intuitive reflection of the weld size of the combined area, as well as the final performance of the process parameters [131]. For polymer–metal hybrid structures, a lap joint is the most common joint mode. The mechanical properties of polymer–metal hybrid joints, including tensile strength, shear strength, peel strength, and fatigue strength, were obtained through tensile shear testing, peel testing, and fatigue testing, respectively [11,135,136]. The weld strength of the lap joint is determined by the lap tensile shear test, and existing studies have generally used tensile shear testing to characterize the bond strength of the joint [11]. Figure 13 shows the schematic diagram of tensile shear, U-Peel, and fatigue tests.

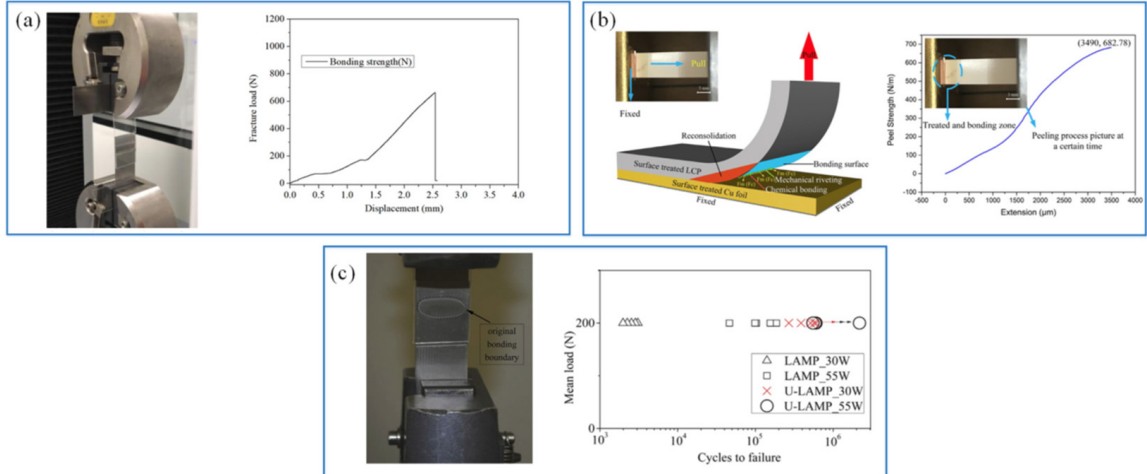

**Figure 13.** The mechanical properties of polymer–metal joints: (**a**) tensile shear strength, (**b**) U-Peel (UP) test, (**c**) fatigue test. Reprinted with permission from [11]. Copyright 2022, Elsevier.

### 3.3.2. Failure Mode

Failure analysis plays a very important role in the manufacture of mechanical components. In laser joining of polymer and metal, the failure mode is an intuitive criterion

for the bearing capacity and energy absorption capacity of joints, supplying an index to evaluate the mechanical properties of hybrid joints. In addition, failure behavior is closely related to the weld morphology [113]. Characterization in terms of failure modes can help to formulate reasonable process parameters and improve the adaptability of applications. Generally, failure modes include interface failure mode, substrate failure mode, and mixed failure mode [137]. Interface failure means that the strength of the joint is significantly less than the strength of the base metal, and the interface of the joint immediately fails. If the strength of the joint is greater than the strength of the base metal, the base metal breaks under tension, the joint does not fail, and the bond is still intact, a process known as substrate failure. Hybrid failure refers to simultaneous interface failure and substrate failure at the bonding interface. Substrate failure is the most frequently encountered type of failure during experimental design. Table 2 shows the typical failure modes of polymer–metal lap joints.

**Table 2.** Typical failure modes of polymer–metal lap joints. Reprinted with permission from [137]. Copyright 2010, Elsevier.

| Mechanism | Description | Appearance | Inference | Details |
|---|---|---|---|---|
| Interfacial | Interfacial failure | | Bond strength $\ll$ substrate strength | Interfacial failure at bond. Least desirable result. |
| Substrate (Type I) | Bulk substrate failure | | Bond strength $\gg$ substrate strength | Substrate results in tensile yield and breakage. This implies that the joint is as strong as possible, and the substrate will fail first. Most desirable result. |
| Substrate (Type II) | Near interfacial substrate failure | | Bond strength $>$ substrate strength | Failure within the substrate, but near the interracial region and the bond remains intact. |
| Mixed (Type I) | Substrate and interfacial | | Bond strength $\approx$ substrate strength | Failure partially within the substrate and at the bond interface. |
| Mixed (Type II) | Interfacial and some substrate | | Bond strength $<$ substrate strength | Failure mostly interfacial, but with some substrate failure, e.g., in the form of plastic deformation. |

*3.4. Defect Characteristics*

In the laser joining of metal–polymer hybrid structures, the forming parts are prone to defects. The occurrence of welding defects directly reflects the selection of process parameters, and is an intuitive representation of forming quality. In laser joining of metals and polymers, common forming defects are bubbles (pores) [114,115,138], lack of fusion [113], discoloration [113], cracking [114,136], deflection [139], etc. Pores are caused by thermal decomposition or cooling shrinkage of the polymer in the bonding zone, and this process is hard to completely avoid. Lack of fusion is caused by insufficient welding energy, so that the polymer does not melt sufficiently in contact with the metal to form an effective joint. Discoloration is due to oxidation degradation caused by contact between the polymer and the air during the laser joining process, which causes polymer discoloration (yellowing, blackening) and even crack defects, and eventually leads to a sharp decline in the mechanical properties of the bonding zone [134,140]. Cracks are the result of high stress during the cooling of the weld pool. Most of the cracks appear where the metal has been subject to direct laser irradiation, or in the pores in the bonding zone, then spread through the bonding zone [114,116,141]. Cracking may also refer to the disconnection of the metal from the polymer during fatigue testing [136]. Deflection is caused by welding-induced deflection of sheet metal, and is closely related to process parameters. The increase in the deflection angle is significantly related to the deterioration of joint strength. This critical defect can cause significant spallation stress, resulting in the failure of welds during post-processing or cooling [139].

From the list of possible defects, pore defects are almost unavoidable. Many researchers have explored the generation and causes of bubbles in the connection process.

Bubbles can be divided into two categories [11,114,142]. The type I pore has large bubbles with regular shapes and smooth inner surfaces, which are mainly distributed near the interface towards the center of the fusion zone between polymer and metal. The reason for this is that the melting point of polymer is low and the bonding interface heat is high, and the gas generated by the thermal decomposition of the polymer does not have enough time to escape. The type II pore involves the shrinking of the pore due to cooling, and the contraction of the gap between the polymer and the metal during melting and re-solidification. Type II pores are small, distributed at a certain distance from the bonding interface, with irregular shapes, rough inner walls, and are often accompanied by small cracks and groove defects. In addition, type I pores usually occur under conditions of thermal decomposition caused by high heat input, while type II pores occur under all heat input conditions.

In polymer–metal joints, the formation, size, number, distribution, and morphology of bubbles have important effects on joint strength and quality. Miyashita et al. [143] showed that bubbles with a suitably sized bonding zone could produce high pressure to promote the combination of molten polymer and stainless steel, and when bubbles exceeded a certain size, the effective contact area of the joint became smaller, leading to a reduction of joint strength. Lambiase et al. [115] demonstrated that in the bonding zone, as the size of bubbles in the central area increased and their distribution became more concentrated, tunnel defects gradually formed in the central area, resulting in a central hollow that reduced the effective connection area of the weld and decreased the joint strength. Jung et al. [120] used Q mass spectrometry to detect submillimeter bubbles generated in the polymer melting zone in the joint of CFRP and stainless steel. The results showed that the chemical components in the bubbles included a series of nitrogen and hydrogen compounds produced by the pyrolysis of CFRP, along with nitrogen in the air, explaining the formation of bubbles in the laser bonding of polymer–metal hybrid joints. Cheon et al. [144] showed that in the laser joining of metal and polymer, the strength of hybrid joints was closely related to the polymer structure, bubble composition, and expansion pressure. Therefore, the formation of bubbles is inevitable although their shapes are different, and the distribution of bubbles has different effects on the bonding strength. By quantifying the bubbles (e.g., size, distribution, etc.), the forming quality can be characterized, and the process parameters can be optimized. Lambiase and Genna [115], studying LAMP between AISI304 aluminum plate and polycarbonate plate in order to determine the influence of processing conditions on the binding quality, extracted a polymer surface brightness histogram as a characterization benchmark, based on the difference of brightness in the bubble region of the bonding area. The relationship between the process parameters and the bubble distribution and forming quality was assessed, as shown in Figure 14. Zhou et al. [145] studied the laser joining of CFRP and steel after surface treatment. By extracting the pixel values of the pores and the joint area in the joint cross section, they calculated relative shrinkage porosity. With porosity as a feature and minimization as a goal, they explored the effect of surface treatment on the inhibition of contractile pores in CFRP–steel. The effects of morphological parameters (protrusion height and protrusion density) on shrinkage porosity were revealed. Schricker et al. [146] used a new type of semi-profile device to directly record the time-dependence of bubble formation in the connection zone. Based on the bubble characteristics, they characterized the relationship between the temperature distribution in the melt zone and bubble defects in the connection process, which strengthened the understanding of the bonding mechanism between polymers and metals.

The effect of bubble formation on joint quality and strength during laser joining of metal and polymers remains controversial. Researchers have claimed that the pressure generated by bubble formation helps the fused polymer adhere to the metal surface, and is crucial for the success of the bonding operation [11,102,114,147]. However, the bubble morphology is different under different laser welding methods and the strength of the connection action also varies. Wahba et al. [148] conducted a comparative study on CJ and LTJ of magnesium alloy and PET. The results showed a difference in the morphology of bubbles between the two welding methods, and a discrete distribution of bubbles in

the interface of the joint was more conducive to the connection between metal and plastic than mesh bubbles in the interface of the transmission welding. Thus, the strength of the joint produced by thermal conduction welding was greater than that resulting from transmission welding. However, Schricker et al. [146] reported a correlation between the origin of bubbles and the dryness of polymers. The results showed that in the absorbent polymer, bubbles were formed in the boundary layer due to the evaporation of water, the effect of the artificial core, or the presence of gas particles in the molten material, and the randomly formed bubbles in the melting layer moved to the boundary layer, meaning the formation of bubbles had a positive effect on the joint connection. For non-absorbent polymers, bubbles from the thermal degradation of polymers originated in the interface and the polymer melt in the process of solidification zone migration. Based on the thermal degradation of air bubbles, the push or extrusion of molten material produced by the so-called positive influence on the metal surface was not established to any great extent. In addition, the generation of bubbles not only reduces the effective bonding area, but also increases stress concentration there, inducing the joint to fail in the pore area rather than at the bond interface, thus affecting the overall forming performance [137,149,150]. More comprehensive studies are needed to define the optimization direction and the appropriate interval when pore defects are used as forming indexes.

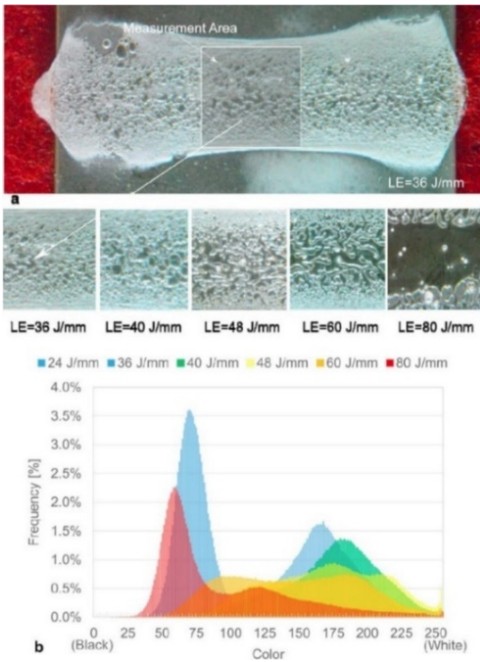

**Figure 14.** Analysis of weld quality: (**a**) macrographs of the weld regions, (**b**) histogram of brightness. Reprinted with permission from [115]. Copyright 2016, Elsevier.

In summary, the quantification of defects allows efficient and intuitive evaluation of the quality of the formation, followed by optimization of the process parameters. As the principle defect, the quantification of bubble (pore) defects can be used as a tool to explore the relationship between bubble (pore) and process parameters, effective bonding area, and bonding strength. However, there remains a lack of data to achieve the optimization objective of bubble (pore) characteristics. In addition, the quantitative characterization of defects such as cracks, discoloration, and lack of fusion has been poorly studied. The characterization of forming quality based on defect characteristics requires further study.

### 3.5. Characteristics of Sensing Signals

Laser bonding of metals and polymers is a process involving the physical, thermal, optical, and chemical properties of materials. Using only visual characteristics, it is difficult to evaluate comprehensively the forming quality after welding, and the characterization

process is time-consuming and destructive to the samples. Therefore, by extracting the sound, light, and heat signals of the welding process, the forming characteristics can be supplemented and online monitoring of the welding process can be used to achieve efficient comprehensive multi-information evaluation of the forming quality.

Thermal characteristics are expressed by thermodynamic signals that encompass the temperature and heat distribution of the molten pool, which can be used to describe thermal conduction, convection, heat radiation, and other phenomena involved in the process of material melting, and can also measure the degree of polymer melting in the bonding zone and predict the forming quality of the bonding [140,151–154]. In the laser joining of AA6082 aluminum alloy and PA66 plastic, Schricker et al. [155] used a K-type thermocouple to record the temperature–time curves under different parameters, described the physical change process of polymer materials at temperature-change points, and evaluated the rationality of process parameters. Huang et al. [133] used a thermocouple to extract the bottom surface temperature of stainless steel. Through temperature analysis, it provided information relevant to polymer melting, bubble formation, and discoloration, and to evaluate indirectly the influence of process parameters on joint quality. In their study of LTJ of polycarbonate and AISI304 stainless steel, Lambiase and Genna [115] used an infrared camera to measure the temperature of the metal surface bonding zone without PC. Based on the temperature characteristics, they analyzed the influence of process parameters on the temperature distribution and development of bond defects, and optimized the process parameter window of the AISI304-PC joint. In addition, in the CJ of PEEK-Al-Mg alloy, the author used an infrared thermal imager to measure the irradiated temperature of the aluminum surface, through analysis of the aluminum alloy surface temperature evolution and distribution under different process parameters. The surface temperature, porosity defects, the metal surface structural parameters, the bonding area size, and the strength of the relationship were given [156]. An infrared camera was used to record the bottom surface temperature of AA5053 aluminum alloy without PVC polymer under CJ, to characterize the thermal field and thermal history of the bonding zone, and then evaluate the influence of technological conditions on the morphology and strength of the joint [157]. Although these measurements did not provide accurate values for the temperature field development during LAMP, they provide a viable evaluation tool for determining a reasonable temperature window for optimization of the process.

Thermal processes and mechanical strains during metal–polymer welding can be recorded by optical fiber sensors. Changes caused by temperature and mechanical strain lead to changes in spectral signals. By evaluating the spectral response signals collected in the bonding zone, the interface temperature changes and residual strain during welding can be captured to evaluate the bonding quality [158]. Wang et al. [159] characterized forming quality by collecting the laser reflection radiation intensity signal during LTJ of metal and polymer, explored the relationship between process parameters and forming characteristics, and developed a laser connection system between polyethylene tereph-thalate plastic and titanium, with controllable heat input. Schmitt et al. [160] used OCT technology to identify and quantify joint geometry, bubbles, breaks, and gaps, based on coherent optical information feedback from samples of the LTJ of metal–polymer, then characterized the forming quality and integrated it into an adaptive online control, to help better understand the formation of bubbles and defects in the connection.

During laser welding, phenomena arising from the interaction of the material and the laser emission, manifesting as radiation, acoustic emission, and electro-magnetic emission, can be monitored and evaluated by various sensing techniques [161–165]. The types of radiation most frequently monitored in conventional laser processing are back-reflected laser radiation, plasma- or metal-vapor-induced radiation, and thermal radiation [166–169]. In addition, certain nondestructive testing techniques can be used to characterize the quality of the formed joint, such as ultrasonic, magneto-optical, ray, eddy current techniques, and others [170–172]. However, laser joining of metal to plastic is different from traditional processes such as transmission welding, where there is no pool of molten metal and no

splash on the surface of the material. Meanwhile, the magnetization ability and radiation absorption energy of polymers are limited. Conventional metal-welding monitoring and feature extraction methods are not suitable for the forming characterization of metal and plastic joints. Therefore, rational selection of the sensing signal and construction of the relationship model between the signal and the forming quality are the keys to good forming, and are the effective foundation for the intelligent development of laser-connected metal and polymer, including monitoring, detection, optimization, and regulation of forming quality.

In summary, the main quality characterization indexes for the laser joining of metal and polymer include the weld characteristics of the metal surface, weld characteristics of the bonding zone, mechanical properties, defect characteristics, and characteristics of the sensing signal. Figure 15 summarizes the forming quality characterization of laser joining between metal and polymer. The weld characteristics of the metal surface only appear in CJ (namely LAMP), directly reflecting the heat input in the processing process. The size of the weld should be minimized to ensure efficient bonding. The weld characteristics of the bonding zone, mechanical properties, and defect characteristics directly characterize the joint morphology, bonding strength, and forming quality, representing the key manifestation of the comprehensive performance of hybrid structures. The characteristics of sensing signals are based on light, sound, heat, and other signals, to complement the above representation of forming quality. Through the reasonable characterization of component forming quality and the analysis of the corresponding requirements, the forming effect can be directly evaluated, which provides reference for the subsequent accurate control of forming quality and process optimization.

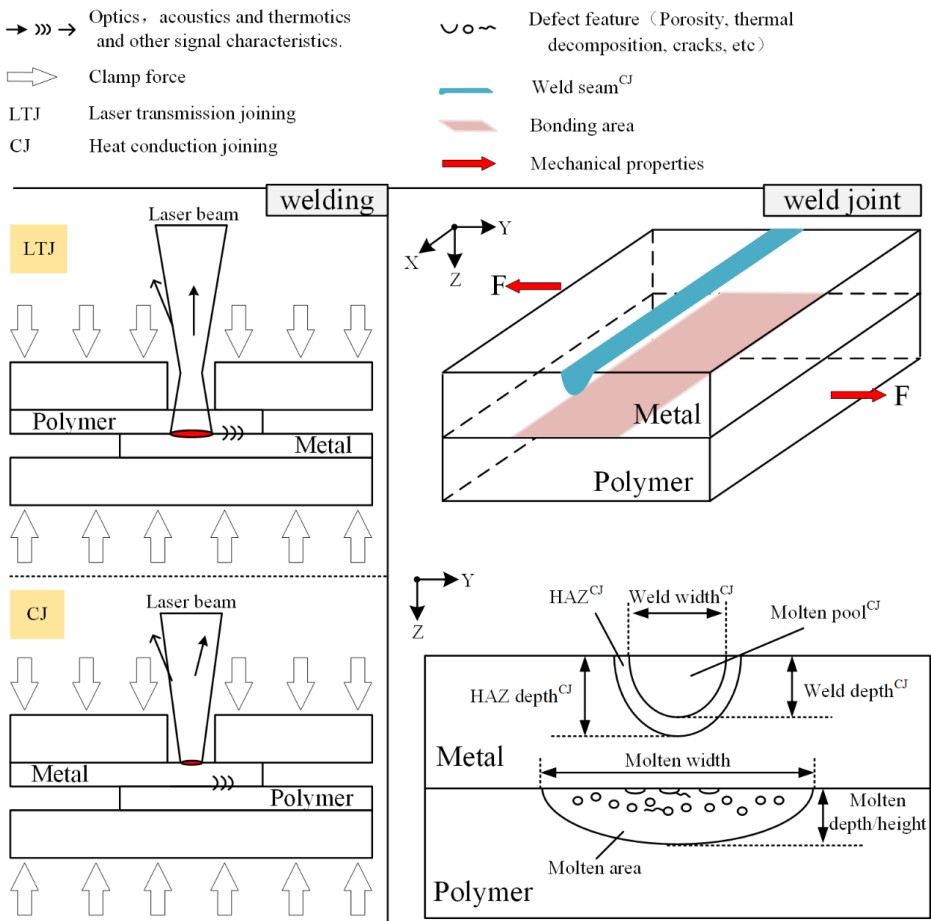

**Figure 15.** The forming quality characterization of laser joining of metal and polymer.

## 4. Forming Quality Control of Metal–Polymer Laser Joining

Laser joining of metal and polymer is a closely related process involving physical, chemical, thermal, and optical characteristics, among others. To obtain hybrid structures with high forming quality, it is necessary to regulate the forming quality. At present, the control methods of metal–polymer laser joining process include mainly surface treatment of the bonding interface, process control, and process parameter optimization.

### 4.1. Surface Treatment of Bonding Interface

In laser joining of metals and polymers, interface bonding depends mainly on three bonding mechanisms: physical, mechanical and chemical. Depending on the bonding mechanism, a variety of methods can be used to treat the bonding interface, enhance the bonding strength, and improve the mechanical properties of the joint. However, physical bonding is weak and does not play a dominant role in the performance of joints between thermoplastic composites and metal heterostructures, so strengthened mechanical bonding and chemical bonding are the main ways to improve the joints' mechanical properties [12].

#### 4.1.1. Microstructure Treatment of Metal Surface

In the bonding process of metals and polymers, the mechanical bonding mechanism refers to the mechanical riveting effect of metals and polymers through a microscopic linkage structure, followed by the realization of a tight connection, essentially through friction forces. After structured surface treatment of metal and polymer in the laser-joining process, the molten polymer under the action of bubble pressure and clamping force is fully activated to fill the pits in the metal surface, such as are caused after quick cooling, polymer solidification, and formation of mechanical interlocking structures, so the anchoring effect and joint bonding strength can be improved [11,13,173]. In addition, the structure has little effect on the surface morphology of the joint, but greatly affects the strength of the joint [148]. By introducing surface microstructure treatment, the state of the metal surface changes significantly, the wetness and roughness of the metal surface increase, the spreading ability of the molten polymer is significantly improved, and so too is the mechanical chimeric effect of the interface [174,175]. Furthermore, changes in surface microstructure can change the heat conduction process at the interface, reduce the heat accumulation at the interface, and inhibit the formation of shrinkage cavities [145].

Metal surface microstructure treatment includes chemical treatment, mechanical treatment, laser treatment, etc. Among the various methods, chemical treatments such as pickling and alkali washing have difficulty achieving accurate control of the surface microstructure, and pollute the environment and the sample surface so that post-treatment is required [12,108]. Mechanical treatment is mainly performed through sandpaper grinding, shot peening, milling, and other methods to control the surface quality. Surface treatment efficiency is high, process pollution is small, and product quality is guaranteed. However, it is difficult to control the surface structure during sandpaper grinding or shot peening, the milling process is affected by the size and shape of the tool, and the microstructure size is usually at the submillimeter level, which limits the effect on the improvement of bond strength [176]. Laser treatment uses lasers to ablate specific microstructures on the metal surface; the processing process is convenient, and the microstructure size is at the micrometer to millimeter level. Through appropriate selection of laser process parameters, scanning strategy, and laser type, a variety of morphologies, sizes and characteristics of microstructure can be obtained, and have been widely studied [12].

In the laser-treated metal surface, the strength of mechanical anchorage is closely related to the basic microstructure parameters. Suitable microstructure morphology, size, and distribution can effectively improve the interface morphology, strength, and forming quality. Rodríguez-Vidal et al. [177] compared the effect of linear microstructure groove angle and groove spacing on joint strength in the laser joining of low-alloy steel (HC420LA) and PA6-GF30. The results showed that the groove spacing was the main factor affecting joint strength, and the shear strength of the joint was inversely proportional to the groove

spacing. In addition, the effects of different depths and widths of microstructure on the binding strength were investigated. Experiments showed that the aspect ratio of depth to width could improve the bond strength within a certain range, but it had little effect on the bond strength beyond this range [178]. Amend et al. [179] prepared pits and grooves of different depths on the surface of 5182 aluminum alloy, and found that with the increase of microstructure depth, the strength of metal–polymer hybrid joints increased. However, when the depth continued to increase, it was difficult to fill due to the limited molten polymer, which led to a decrease in the speed of strengthening the joint. With the less complete filling, the void in the bonding zone increased, resulting in a decrease in strength. Jiao et al. [180] prepared a network micro-texture with various spacings on the surface of A7075 aluminum alloy, and demonstrated that when the spacing between the microstructures was within a reasonable range, the combination of interface morphology and quality was at its best. In addition, when different laser scanning strategies were used on the metal surface, different microstructure morphologies were obtained, such as conical [181], grid slot [179], rhombus [182], or porous [183]. However, in the laser-structured metal surface, metal melt inevitably experienced rapid cooling or multiple heating and melting processes, causing metal surface heat defects such as cracks, holes, and edging, and the area's microstructure and mechanical properties were reduced, affecting the hybrid joint connection strength and the forming quality [182,184]. Therefore, in the process of micro-structuring the metal surface, in addition to considering the basic characteristics of the surface microstructure parameters, the structural process parameters need to be correctly regulated to improve the overall forming quality.

4.1.2. Chemical Treatment of the Bonding Interface

In the bonding mechanism of hybrid joints, chemical bonding refers to the formation of new chemical bonds between the metal surface and the surface-active functional groups of atomic polymers, forming a connection due to the mutual migration of charges. In addition, due to the lack of active functional groups on the surface of some polymers, such as PP-SUS304 [185], it is difficult to form effective bonding between metal and polymer at the interface. Therefore, by modifying the matrix surface of the composite material or metal surface before welding, the chemical bond between them can be effectively enhanced.

In terms of surface state treatment of composite materials, Arial et al. [138] used UV-ozone and plasma to pretreat and modify COP (cyclic olefin polymer) surfaces, and found that after surface treatment, C-O bonds and C=O bonds on the COP surface increased, new COO bonds were formed, the proportion of oxygen functional groups increased, and the surface energy also increased. The interfacial interaction between the oxygen functional group and the $Cr_2O_3$ oxide film of SUS304 increased, and the strength of the laser-connected SUS304-COP joint increased. Zhang et al. [186] used acrylic acid as a graft monomer to graft the surfaces of carbon-fiber-reinforced composites with UV light, resulting in a large number of O-C=O and C=O bonds on the surfaces of the composites, which promoted the formation of "AL-C" and "Al-O-C" chemical bonds at the interfaces between the composites and the aluminum alloy. At the same time, the wetting angle of the composite surface was reduced and the interface bonding strength was improved. In addition, as well as considering the wetting angle, this study shown that the UV grafting process was affected by the UV irradiation time, which determined the type and content of chemical bonds on the surface of the composite. However, there has been little work on polymer surface modification and no systematic study of bond strengthening between modified polymers and metals.

In the treatment of the metal surface state, coatings, thermal oxidations, or the introduction of functional groups can be used to change the metal surface state, in order to achieve chemical connections and strengthen the joint quality. Jung et al. [187] placed a galvanized steel plate into an air furnace for heating and oxidation, and the joint strength of laser joining of the acrylonitrile–butadiene–styrene copolymer (ABS) and galvanized steel plate was improved. The authors speculated that ZnO layer was generated on the

surface of the galvanized steel plate, and this oxide easily formed a chemical bond with ABS. Arkhurst et al. [141] applied heat treatment to magnesium alloy before welding, in their study of laser joining between magnesium alloy and carbon-fiber-reinforced plastics. The results shown that after annealing and heat treatment, an oxide layer formed on the surface of the magnesium alloy, and the thermal oxidation effect could effectively suppress bubble defects. The bonding strength was doubled by simultaneously enhancing the mechanical linkage effect and the chemical bonding process. Tan et al. [149] electroplated a Cr layer of certain thickness on the surface of low carbon steel, and found that a chemical bond of Cr-O-PA6T was formed on the interface, which significantly improved the joint strength of steel–CFRP. In addition to surface coating, thermal oxidation, and other pretreatment methods to promote chemical modification of metal surface bonding, researchers haave used chemical reagents and functional group preparation methods to provide functional groups to the metal surface, to form chemical and hydrogen bonds with the molten thermoplastic composite material at high temperature, to achieve the purpose of chemical control. Pan et al. [188] used a Schiff base complex to improve the wetness of the TC4 surface, promoting the appearance of more active groups at the interface, inducing the formation of chemical bonds with polymer materials at the interface, and further improving the bonding strength between TC4 titanium alloy and polyether ether ketone. Hino et al. [189] added a styrene block copolymer (SBC) containing a carboxyl group (-COOH) and an amino group (-NH$_2$) to the interface, in order to improve the laser joint strength of aluminum alloy and PP, which promoted the formation of hydrogen bonds at the interface and further improved the bonding strength. The above control methods generally change only in the chemical state of the metal surface. Anodic oxidation and micro-arc oxidation surface treatment change the chemical state of the metal surface, forming a layer of micro and nano scale apertures in the porous morphology of the metal oxide film, promoting the adsorption of chemical functional groups on the metal surface, further providing the foundation for chemical control of morphology, and to a certain extent improving the interfacial mechanical bonding effect. Yusof et al. [123] studied laser joining between anodized A5052 aluminum alloy and polyethylene terephthalate (PET). It was found that the porous films obtained on the surface of the aluminum alloy by anodized oxidation improved the wettability of molten resin on the surface of the metal matrix, and significantly improved the mechanical bonding strength. The formation of chemical bond was further promoted, and the bonding strength of the interface was enhanced. Zhang et al. [190] formed alumina with nano-porous structure on the surface of aluminum alloy by rationally selecting the anodizing process parameters on the surface of the aluminum alloy, which greatly enhanced mechanical anchorage while forming an Al-O-PA6 chemical bond at the bonding interface and greatly improving the joint strength (from 5.3 MPa to over 40 MPa). Micro-arc oxidation is based on ordinary anodic oxidation, using a dedicated micro-arc oxidation power supply voltage on the workpiece, involving the interaction between the metal and the surface electrolyte solution to produce a ceramic membrane on the metal surface, under the influence of factors such as high temperature and electric field, and to achieve the purpose of surface strengthening and surface modification of the workpiece. Pan et al. [191,192] studied the effect of micro-arc oxidation treatment on the connection between magnesium alloy and CFRP, and found that the pre-treated magnesium alloy surface formed a porous oxide film, which improved the wettability of melting resin on the magnesium alloy surface. The mechanical bonding effect was significant, the contact area between the two was increased, and the bond strength was significantly improved. However, there was no comparison of bonding mechanisms before and after micro-arc oxidation, and the neither the change of chemical bonding at the interface and nor the analysis of the bonding mechanism were discussed.

Considering the characteristics of each bonding mechanism, the surface microstructure and surface modification are combined to realize the joint action of multiple bonding mechanisms, and the forming quality of the bonding interface can be further regulated. Su [193] used laser micro-texture and the micro-arc oxidation–silane coupling composite regulation process to treat TC4 surface morphology, optimized the pretreatment process

parameters combined with the response surface method, and performed laser-joining experiments with CFRTP. The results showed that the composite control method further improved the adsorption capacity of the TC4 surface to melted CFRTP at high temperatures, realized the complete filling of micro-texture and micro-arc oxidation film, alleviated the interfacial stress concentration, and changed the surface state of TC4. It could promote the formation of Ti-C, Ti-O, and other new chemical bonds between TC4 surface active elements and C and O elements in CFRTP at high temperature, and improved the adsorption capacity of hydroxyl (-OH) on the TC4 surface. It promoted the introduction of interface amino functional groups and realized hydrogen bond bonding with a carbonyl group (C=O, R-CO-R) and ether bond (R-O-R) at high temperatures, which significantly improved the bonding strength of the joint from three aspects: mechanical chimerism, chemical bonding, and hydrogen bonding.

Summarizing the above, Figure 16 shows a schematic diagram of surface control in the bonding interface. Laser treatment technology can control the size and distribution of microstructures over a wide range, making it an ideal method for fabricating microstructures, and is beneficial for inhibiting the formation of interface defects. UV light-grafting technology can introduce specific functional groups onto the resin surface, but how to ensure controllability of the process needs further investigation. The metal anodization method has been tentatively shown to be a viable interface-strengthening method worthy of further investigation. However, the process of preparing oxidized film on anodized surfaces is complicated, the film formation time is short, and it is difficult to realize film production on a large specimen surface, so its industrial production and application are limited. The industrial applications of micro-arc oxidation are mature and can be employed to fabricate porous films on large metallic surfaces. However, there has been very little research on laser joining to strengthen hybrid joints [194]. The synergistic effects of mechanical strengthening and chemical bond strengthening can further improve the mechanical properties of joints, but this method has difficulty realizing the machining of large size components due to the limitation of machining conditions.

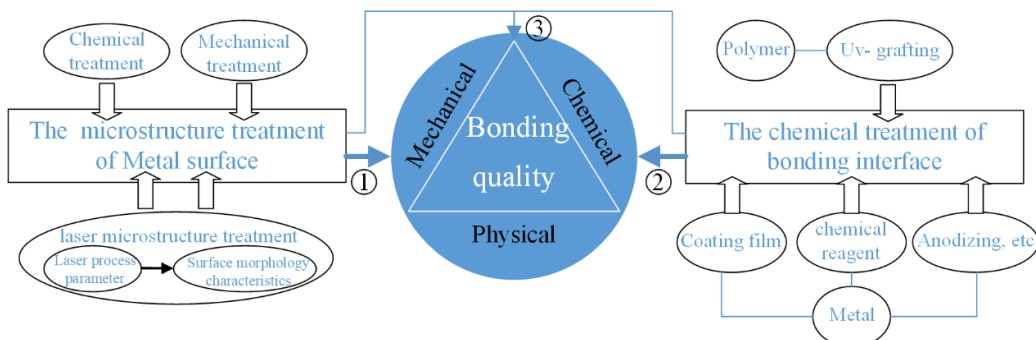

**Figure 16.** Schematic diagram of surface control at the bonding interface.

*4.2. Process Control of Laser Bonding Metal to Polymer Process*

In laser joining between metals and polymers, the quality of the joint is closely related to polymer melting, spreading, and cooling solidification during the forming process. By applying auxiliary equipment, beam shaping, adding intermediate layers, and introducing high thermal conductivity blocks, the solidification condition, fluidity, and spreading ability of the molten pool in the forming process can be improved, to inhibit the formation of defects in the forming process, so the forming quality can be well regulated, as shown in Figure 17.

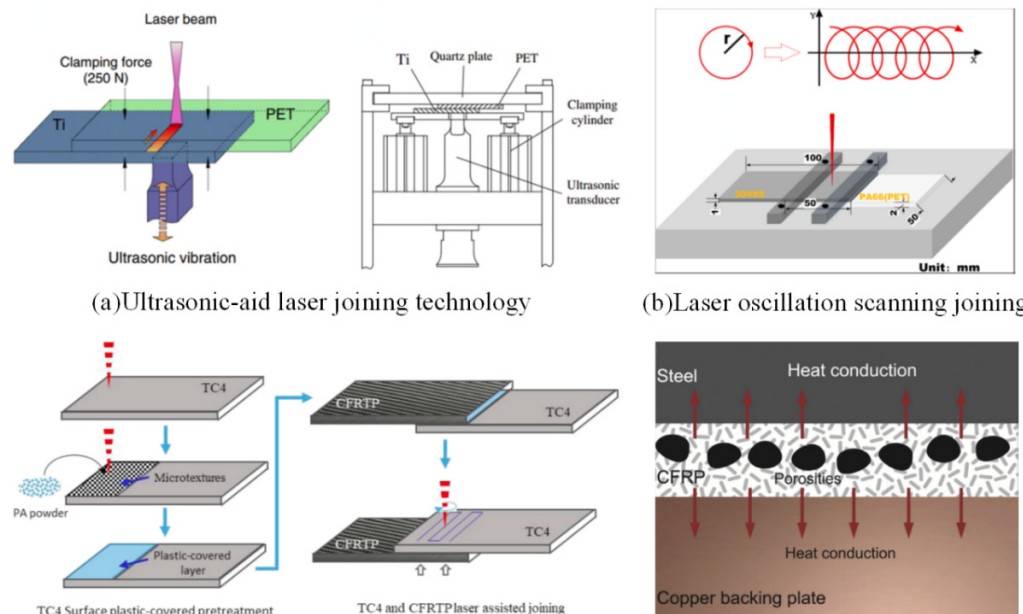

**Figure 17.** Process control method for laser joining of metal and polymer. Reprinted with permission from [103,114,195,196]. Copyright 2016, 2021, 2021, 2014, Elsevier.

In the laser joining of metals and polymers, the process of adding auxiliary equipment to improve quality mainly refers to added ultrasonic vibration assistance. Chen et al. [103] proposed a new method of laser connection between metal and plastic, assisted by ultrasonic vibration. The results showed that ultrasonic vibration promoted the chemical reaction between PET and Ti metal, resulting in high strength $TiO_2$, $Ti_2O_3$, Ti-O, and Ti-C bonds. Compared with the original joint, a thicker chemical binding interface was formed. In addition, the combined effect of ultrasonic vibration and thermal field affected the bubble movement trajectory and promoted bubble escape from the joint, which significantly reduced bubble defects at the joint, and finally greatly improved the joint strength and fatigue performance. The strength of the UAL (ultrasonic-aid laser joining) joint was at least four times higher than that of the LAMP joint, and the overall forming quality was improved [135,197,198]. Liu et al. [199] synchronously applied ultrasonic vibration for auxiliary welding during the process of laser conduction joining between surface microstructure 316L stainless steel and PET. The results showed that in ultrasonic vibration, the acoustic flow effect improved the fluidity of the PET solution, making it easier to fill the texture, and ultrasonic cavitation changed the morphology of bubbles, reduced the bubble rate in the joint, increased the contact area, and produced higher pressure. Under the combined action of surface texture and ultrasound, the joint strength could reach up to 45 MPa, which was 86.5% of the strength of the PET plastic base material.

In recent years, with the development of laser technology, beam shaping has been more frequently applied during dissimilar material laser welding, including line laser welding [148], spot laser beam welding [200,201], laser oscillatory welding [195], etc. Among these, laser oscillation welding has great potential, because it supplies more diverse energy distribution, can realize the movement of the laser beam according to demand, so that the welding energy control is more accurate, and can greatly improve the polymer–metal joint weld forming quality [11,202]. Researching the laser welding of carbon fiber reinforced polymer (CFRTP) and aluminum (Al) alloy, Jiao et al. [118] introduced a high-speed rotating laser welding technology, which controlled the moving path of the laser beam by use of a scanning galvanoscope, to enable circular oscillatory welding. The results shown that this welding method could significantly reduce heat accumulation and pore defects during welding process, and improved the mechanical properties of joints. Gao et al. [195] used

laser oscillation technology to connect PA66 plastic and 304 stainless steel, and studied the influence of process parameters such as beam scanning radius and defocus on the performance and mechanical properties of the joint. The results showed that compared with the traditional LAMP, the oscillating laser beam could effectively homogenize the heat distribution and control the width and bubble distribution of the joint, while the tensile strength of the joint was increased by 28.9%. Hao et al. [203] studied the laser oscillation transmission connection of PET-304 stainless steel and suggested that the laser oscillation behavior could eliminate thermal cracking defects, homogenize the weld morphology, and improve the forming quality. Bu et al. [204] used a laser with a swing length of 2 mm to connect CFRTP with 6061 aluminum alloy, and found that the laser swing process was beneficial for obtaining a more uniform temperature field and a larger bonding area. It has been demonstrated that the laser oscillation technique outperforms the LAMP technique, which not only broadens the tuning window of the laser connection process, but also represents a clear advantage in terms of energy regulation. In addition, several new hybrid welding techniques, such as adjustable ring mode (ARM) and dual laser beam, are expected to be applied in the production of high quality hybrid structure joints [11,205]. However, most studies have focused on improving the mechanical properties of hybrid structures.

Joint quality can also be enhanced by adding an intermediate layer of appropriate thickness onto the joint surface. Before LAMP of CFRTP and TC4 alloys, Jiao et al. [196] applied PA powder cladding on the surface of TC4 after microstructure treatment, to increase the melting quantity of the bonding interface resin and improve the bonding quality. In the laser joining of aluminum alloy–CFRP and 304 stainless steel–CFRP, researchers added an intermediate resin layer to increase the amount of fused resin in the joint surface, thus avoiding the formation of defects. The shear strength of the aluminum alloy and CFRP joint increased from 15.8 MPa to 37.5 MPa. The shear strength of 304 stainless steel and CFRP joint was 116% higher than that of the original joint [180,206,207].

Considering the difference in thermal expansion coefficient and thermal conductivity between polymer and metal, high thermal conductivity sheets can be introduced to increase the polymer's thermal conductivity and improve the interface bonding quality. Tan et al. [114] investigated the effects of addition of copper plates onto the bonding joint on the reverse (polymer side) of the welding during LAMP between CFRP and stainless steel. The results showed that the addition of a copper plate enhanced the cooling effect and could effectively suppress shrinkage and improve the quality of the formation.

*4.3. Process Parameter Optimization*

The process of laser joining between metals and polymers is complex and often involves multiple physical fields. The forming quality of a hybrid structure is directly related to laser power, welding speed, defocus, clamping pressure, and other process parameters [11,131,208]. Among these, the exciting power and welding speed are directly related to the heat input. By controlling the heat input during the connection, thermal damage to the polymer matrix caused by thermal laser action can be minimized while ensuring the connection, and a reliable interface connection can be formed [11]. Defocus is directly related to the spot diameter. A larger spot size can increase the width of the bonding surface, reduce the heat input energy density, and reduce the thermal damage to the composite material [206]. The clamping force is an important parameter for ensuring a tight connection, and directly affects the bubble characteristic parameters and bonding quality of the lap interface. A greater clamping pressure can promote the flow of molten polymer, increase the contact area, and reduce the occurrence of shrinkage caused by the difference of thermal expansion coefficient [143,209,210]. When pulsed laser welding is used, the forming quality is also related to peak power, pulse frequency, pulse waveform, and pulse width [113,129]. The type of laser, wavelength, scanning strategy, optical properties of materials, size of composites, fiber content, and other variables can affect the forming efficiency and quality to a certain extent [11,104,211–213]. The quality of forming is finally

determined by the common effect of influencing factors. Therefore, the optimization of forming process parameters is the key to controlling forming quality.

As shown in Figure 18, the optimization of process parameters includes single factor methods, mathematical statistics, intelligent algorithms, and numerical models. These mathematical methods can be used to optimize welding parameters efficiently. The single factor method involves carrying out many repetitive experiments while changing a single process parameter, and can be combined with the forming-quality characterization method to obtain the optimal values of process parameters. The whole procedure is intuitive and convenient, but it is costly and time-consuming, so it is difficult to obtain the optimal parameters [116,117]. The optimization method involving mathematical statistics relies on orthogonal experimentation, full factor experimentation, central composite experimentation, and other methods, combined with range analysis, variance analysis, gray correlation degree, and other statistical analysis to obtain the optimization parameters. In the study of PMMA and 304 stainless steel welded by pulsed LAMP, Huang et al. [113,214] carried out experiments with orthogonal and Taguchi designs, respectively. Taking the bonding strength and weld width as forming characteristics, and combined with range and variance analysis, they optimized the process parameters and obtained a suitable forming quality. Lambiase and Genna [114] carried out full factor experiments for laser joining of polycarbonate (PC) and stainless steel AISI304, and analyzed the effects of main process parameters such as scanning speed and laser beam power on temperature distribution, morphology, bubble formation, bonding zone size, and mechanical bonding behavior. The optimal combination of process parameters was obtained with the goal of having the fewest bubble defects, the highest bonding strength, and the largest effective binding area. Tamrin et al. [215] studied laser joining between ceramic and plastic using full factor experiments and grey correlation analysis. The relationship between weld strength, weld width, and Kerf width as a function of laser power, distance, and welding speed was optimized, and the optimal parameter combination was determined. Using mathematical statistics to optimize process parameters has the advantages of intuitive process and simple calculation, but the accuracy and efficiency of this method need to be improved [134].

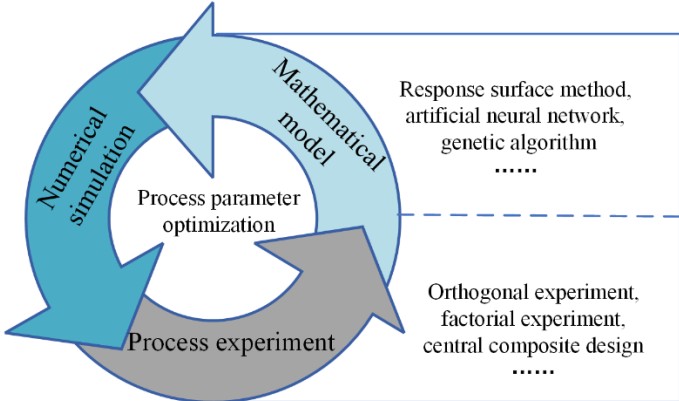

**Figure 18.** Process parameter optimization measures.

Intelligent algorithms can be used to build a mathematical model connecting input and output, bringing the experimental data into the optimization of the process parameters, and predicting the forming quality. Common intelligent algorithms for laser joining between metals and polymers include the response surface method and the neural network algorithm. Hussein et al. [129] used the response surface method (RSM) to study the LTJ and CJ between 304 stainless steel and PMMA. The influence of peak power, pulse duration, pulse repetition rate, scanning speed, and pulse waveform on the joint strength and width of PMMA connected to 304 stainless steel was optimized. The results shown that the prediction model was in good agreement with the experimental results. Al-Sayyad et al. [139] studied the effects of laser welding parameters on the strength and mass of titanium alloy

(TI-6Al-4V)–polyamide (PA6.6) components. The bonding strength and deflection were taken as forming characteristics, and the process window was determined by combining surface morphology analysis and variance analysis. The response surface method (RSM) was applied to construct the mathematical model of process parameters and forming quality, optimize the process parameters, and obtain high quality joints. Acherjee et al. [216] selected shear strength and weld width as forming characteristics, established a nonlinear model between LTJ parameters and output variables using artificial neural networks, and optimized process parameters including laser power, welding speed, distance, and clamping pressure. Meanwhile, multiple regression models were constructed and compared, and it was found that the neural network model had better prediction accuracy. Huang [217] studied the influence of treatment parameters including laser power, welding speed, spot size, and pulse frequency on weld width and shear stress during laser transmission welding of 304 stainless steel and PMMA, and optimized each process parameter by choosing orthogonal experimental data to build a BP-ANN neural network model. The prediction ability of the model was good, the maximum relative error was 11.7%, and the accuracy was less than 12%. In addition, intelligent algorithms such as genetic algorithms, particle swarm optimization algorithms, and their improved algorithms are common methods for optimization of laser welding process parameter for dissimilar materials, and can be applied for the optimization of metal and polymer laser connections [218–220]. However, intelligent optimization algorithms depend on the accuracy of forming process parameters and the precision of the index relationship model. The response surface method is not applicable for the optimization of four or more process parameters, nor is it suitable for investigation and modeling of scenarios with multiple simultaneous targets, while the relational models established by neural networks usually have some shortcomings such as poor interpretation [221,222].

By means of numerical simulation and experimental verification, the temperature field of the joint interface can be studied, the forming quality predicted, and the process parameters optimized. Lambiase et al. [223] used numerical simulation to calculate the temperature field distribution at the interface between composite materials and metals under different process parameters, and verified the simulation results through experiments. The results showed that the temperature field of the interface was affected by the laser power and welding speed, and the numerical simulation method could be used to optimize the process conditions and the shape and size of the laser beam, to reduce the defects in the joint and improve the forming quality. Rodrıguez-Vidal et al. [224] combined experimentation and the finite element method to study the connection process parameters, providing the basis for the selection of interface temperature parameters and significantly reducing experimental trial and error costs. In addition, the combination of numerical simulations and intelligent algorithms allows more comprehensive control of the formation quality. Acherjee et al. [225] studied laser welding of polymers using a combination of finite element method and response surface simulation. First, the finite element method was used to simulate the temperature field and weld size within the welding process, then the accuracy of the model was verified by experiments. The effects of the process parameters on the weld quality were investigated using the response surface method, and the process parameters were then optimized to obtain the desired weld quality. Wang et al. optimized the laser welding parameters of PET thin-film titanium [226] and PET thin-film AISI 316L sheet [227], respectively, by using a combination of numerical model and RSM, and the predicted joint strength values were in good agreement with the experimental results. In terms of optimization and regulation, numerical modeling has received considerable attention because it provides a deeper understanding of process mechanisms, but it is also time-consuming [228–230].

In laser joining of metal and polymer, high quality components can be obtained effectively by optimizing process parameters. Existing studies have mainly focused on joint bonding strength and weld width as the basis for investigating joint shaping quality and optimizing process parameters. However, the forming characterization of metal–polymer

hybrid joints should also involve the surface quality, porosity characteristics, and forming accuracy of welds. In addition, to further promote the application of laser-connected hybrid joints in the automotive field, it is necessary to consider comprehensively the mechanical properties, weight, cost, safety, environmental protection standards, and other features of the base metal and hybrid structure, to optimize the components within the multi-objective framework. Therefore, suitable selection of the forming index, component process parameters, and component performance mathematical model is the key to optimization of process parameters, and is necessary for good control of forming quality and component application.

## 5. Conclusions and Outlook

Metal–polymer hybrid structures have been widely used in the automotive industry due to their excellent bending, impact, and fatigue resistance, as well as their potential for lightweight application. As an efficient processing technology, laser joining of metals and polymers has great application prospects for lightweight automotive manufacturing. However, reduction of automotive weight is a complex multi-objective optimization process, and the research requirements of using laser joining for lightweight automotive components are not clear. In addition, certain problems seriously affect the high quality forming of hybrid structures and the development of the technology; these problems include a small process window, forming defects, and low bonding strength. Therefore, in this paper, the research statuses of certain metal–polymer composite structures for lightweight automotive application were summarized, and the advantages and development of laser-bonding technology were discussed. Furthermore, the characterization indexes for forming quality in the laser joining of metal–polymer hybrid structures were summarized and evaluated, along with the optimization regulation methods.

1. Laser joining of metal and thermoplastic polymer has great application potential for automotive research. To realize the direct use of laser joining metal–polymer hybrid structures in lightweight automotive applications, it is necessary to conduct multi-objective optimization research on components, from the aspects of structural optimization, material optimization, and connection technology selection, based on the performance indexes and functional requirements of selected automotive components.

2. In order to achieve high quality forming of metal and polymer laser joining, the forming quality of the structures can be characterized. The quality characterization indexes for laser joining of metal–polymer mainly include weld characteristics of the metal surface, weld characteristics of the bonding zone, mechanical properties, defect characteristics, and characteristics of sensing signals. In the process of laser connection, a good quality joint should have as few weld characteristics on the metal surface as possible, large weld size at the interface, high mechanical properties, ideal morphology, and low formation of defect characteristics.

3. In the process of metal and polymer laser joining, surface treatment at the bonding interface, process control, and optimization of process parameters are effective means to achieve forming quality control. Hybrid joints with high binding strength, low stress concentration, and high forming quality can be obtained by the combination of metal surface microstructure, chemical modification of the binding interface, and reasonable optimization of process parameters. In addition, the solidification condition, fluidity, and spreading ability of the molten pool can be improved to inhibit the formation of defects during the forming process by applying ultrasonic auxiliary equipment, beam shaping, adding intermediate layers, and other processes control methods, which can lead to good control of the forming quality. Considering the influence of process parameters on forming quality, based on the above quantitative characteristics of forming quality, methods such as mathematical statistics, intelligent algorithms, and numerical simulation can be used to optimize the process parameters.

At present, there has been insufficient in-depth study of the morphology, defects, and costs of metal and polymer laser junctions, and inadequate characterization of their

forming quality. Further studies on these aspects are required. At the same time, in order to gain a deeper understanding of the process mechanisms and parameter effects, and to achieve economic and efficient welding quality, the modeling and regulation of the laser joining process will remain the focus of research. With the development of laser joining technology and intelligent algorithms, based on the existing characterization index and evaluation system of forming quality, clear research goals emerge. These include how to establish the relationship model of process parameters and forming quality, considering the multi-objective requirements of lightweight automotive applications, how to describe accurately the relationship model of process parameters and forming quality, and how to optimize and regulate the process parameters and forming quality by selecting intelligent algorithms. The development of a fast, accurate, and cost-effective laser joining control system with a strategy for lightweight automotive application is a reliable way to achieve the development of hybrid laser-connected structures in the future.

**Author Contributions:** Conceptualization, Z.Z.; funding acquisition, X.G.; project administration, X.G.; investigation, Z.Z.; methodology, Z.Z.; supervision, X.G.; writing—original draft preparation, Z.Z.; writing—review and editing, Z.Z., X.G. and Y.Z. All authors have read and agreed to the published version of the manuscript.

**Funding:** This work was partly supported by the Guangzhou Municipal Special Fund Project for Scientific and Technological Innovation and Development (202002020068).

**Institutional Review Board Statement:** Not applicable.

**Informed Consent Statement:** Not applicable.

**Data Availability Statement:** Not applicable.

**Conflicts of Interest:** No potential conflict of interest is reported by the authors.

## Abbreviations

The following abbreviations are used in this manuscript:

| | |
|---|---|
| CFRP | Carbon Fiber Reinforced Composites |
| HSS | High Strength Steel |
| PMH | Polymer Composite-Metal Hybrid Structure |
| GFRP | Glass Fiber Reinforced Composites |
| GLARE | Glass Fiber Reinforced Aluminum |
| PP | Polypropylene |
| EPDM | Ethylene Propylene Diene Monomer |
| LFT | Long Fiber reinforced Thermoplastics |
| TWB | Tailor Welded Blank |
| TRB | Tailor Rolling Blank |
| FRP | Fiber Reinforced Plastics |
| SMC | Sheet Molding Compound |
| NVH | Noise, Vibration and Harshness |
| BF | Basalt Fiber |
| PLA | Polylactic Acid |
| NFRP | Natural Fiber Reinforced Composites |
| PE | Polyethylene |
| PA | Polyamide |
| PEEK | Polyether Ether Ketone |
| PC | Polycarbonate |
| IM | Injection Molding |
| FDM | Fused Deposition Modeling |
| LTJ | Laser Transmission Joining |
| CJ | Heat Conduction Joining |
| LAMP | Laser Assisted Metal and Plastic |
| CFRTP | Carbon Fiber Reinforced Thermal Plastic |
| PET | Polyethylene Terephthalate |

PMMA    Polymethyl Methacrylate
COP     Cyclic Olefin Polymer
UV      Ultraviolet
ABS     Acrylonitrile Butadiene Styrene Copolymer
UAL     Ultrasonic-aid Laser Joining
ARM     Adjustable Ring Mode
RSM     Response Surface Method

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
