# Peer review of "Research Progress on Characterization and Regulation of Forming Quality in Laser Joining of Metal and Polymer, and Development Trends of Lightweight Automotive Applications"

_metals, doi:10.3390/met12101666_

Round 1
Reviewer 1 Report
The present manuscript summarized the research progress in laser joining of metals and polymers. However, the reviewer is not convinced about the need for this review for the following reasons:
A Scopus survey has revealed at least 5 review articles have been published in the year 2022 itself. Additionally, the same corresponding author already has published a review article on the same topic named " Laser joining technology of polymer-metal hybrid structures - A review" in the journal of manufacturing processes just a month ago. Are the authors making "salami slices" of the same topic? Is not it against the publication ethics?
A good review has a number of telling features: it is worth the reader's time, timely, systematic, well written, focused, and critical. However, the present manuscript is just a summary of a few improvements in the research activities on the current topic without any critical discussion, which ultimately leads to the direction of new research.
This manuscript is written like a book chapter. For example, no logical development of arguments was presented in chapter 2.
2 Metal and polymer hybrid structures for automotive lightweight and laser joining
2.1 Research status of metal-polymer hybrid structures in automotive lightweight
2.1.1 Automotive body and its structural parts
(1) The automotive body frames
As you can see above, not a single statement was made between four consecutive headings. It definitely lost the attention of the reader.
Massive grammatical editing is required. Some examples are below:
Yang et al. 22
Park et al. 24
Yang et al. 27
In addition, the maximum intrusion amount and peak intrusion velocity of the door inner panel decreased and the crashworthiness of the door increased under the condition of vehicle collision 37,.
Fiber reinforced thermoplastics have been widely concerned because of their excellent mechanical properties 57. What is the number 57 here?
region and the bonding ratio of carbon fiber and resin mixture 586 measured by Image J in the bonding region were extracted to study the bonding quality, 587 as shown in Error! Reference source not found..
This kind of mistake appears in almost every sentence throughout the manuscript.
It is just impossible to follow the storyline.
There are no linking sentences between each of the headings. For example, the reader has no idea what will come after the statement "The results indicated that the designed lightweight body not only met the performance requirements but also provided greater torsional stiffness." Each of the headings comes as a surprise.
No logical structure was observed within the headings and subheadings. This is true for each of the chapters from two to four. Just as an example, Chapter 3 was sub-sectioned as below:
3 Forming characterization of metal-polymer hybrid joint
3.1 Weld characteristics of metal surface
3.2 Weld characteristics and quantification in bonding zone
3.3 Mechanical properties
3.4 Defect characteristics
3.5 Characteristics of sensing signal
I cannot find any logical relationship between each of the subsections. How come the Characteristics of sensing signal falls within the heading of Forming characterization of the metal-polymer hybrid joint?
Line 380-381: What do the authors mean by the following statement?
Among them, riveting includes mechanical riveting (riveting without rivet)90-, self‑piercing riveting 94-, friction riveting 97,, friction self-riveting 99,, electromagnetic riveting 101, flow drill screwing 102, friction stir riveting 103104, etc.
What does the author mean by 103104? Also, when the reviewer looked at the reference, it was observed that the authors used a very old reference for self-piercing riveting. At least 4 new review articles were written later on the topic of self-piercing riveting.
Reviewer 2 Report
This review paper has suggested comprehensive information on the metal-polymer hybrid structure joining. However, the author’s argument is ironically obscured due to the too much information. Some sentences are too long (5-7 layers) to read it. It’s better to reduce the less important paragraphs. This paper seems to need more editing service.
1. It is recommended to insert a list of abbreviation
2. Check the reference style used in manuscript
3. Recommend to alter the below title. "Forming” is insufficient word to involve the content.
“3. Forming characterization of metal-polymer hybrid joint” & “4 Forming quality control of metal-polymer laser joining”.
4. Section 3.5 paragraph more appropriate for 4.0 section. This is because sensing signals are usually used to optimize and control the process.
5. When the materials and processes are complexly mixed, inserting tables or schematic diagrams can enhance the reader’s understanding.

Round 2
Reviewer 1 Report
The authors made efforts to improve the manuscript. it is much better now.